# Video-Based Optimal Transport for Feedback-Efficient Offline Preference-Based Reinforcement Learning

**Tung M. Luu** [1]   **Hwanhee Kim** [1]   **Younghwan Lee** [1]   **Chang D. Yoo** [1]

## Abstract

Conveying complex objectives to reinforcement learning (RL) agents often requires meticulous reward engineering. Preference-based RL (PbRL) offers a promising alternative by learning reward functions from human feedback, but its scalability is hindered by high labeling costs. Inspired by advances in Video Foundation Models (ViFMs), we present Video-based Optimal Transport Preference (VOTP), a semi-supervised framework that learns effective reward functions from only a handful of labels. By leveraging optimal transport to align visual trajectories within the rich representation space of ViFMs, VOTP effectively generates high-fidelity pseudo-labels for large amounts of unlabeled data, substantially reducing human supervision. Extensive experiments across locomotion and manipulation benchmarks demonstrate the superiority of VOTP, which outperforms state-of-the-art offline PbRL methods under limited feedback budgets. We also showcase the robustness of VOTP in the presence of visual distractors and validate its utility on real robotic tasks, where it learns meaningful rewards with minimal human input.

## 1. Introduction

Reinforcement learning (RL) has been successful in solving various decision-making tasks when a suitable reward function is available (Mnih et al., 2015; Silver et al., 2017; Haarnoja et al., 2018; Chen et al., 2022b). Yet in many real-world scenarios, reward design remains challenging. Constructing dense and informative rewards often requires extensive instrumentation, such as motion capture systems (Gupta et al., 2016), proprioceptive sensors (Zhu et al., 2019), or tactile sensors (Koenig et al., 2022). Even with such re-

[1]Korea Advanced Institute of Science and Technology (KAIST). Correspondence to: Chang D. Yoo <cd_yoo@kaist.ac.kr>.

*Proceedings of the $43^{rd}$ International Conference on Machine Learning*, Seoul, South Korea. PMLR 306, 2026. Copyright 2026 by the author(s).

sources, reward misspecification can still occur, in which RL agents discover and exploit unintended shortcuts in the reward function (Skalse et al., 2022). In these cases, the reward signal may be maximized, but the resulting behaviors are often undesired or even harmful (Clark & Amodei, 2016; Popov et al., 2017).

Instead of hand-engineering reward functions, many works learn them directly from human data, such as expert demonstrations (Abbeel & Ng, 2004), natural language (Fu et al., 2019), and human feedback (Yuan et al., 2024). Recently, preference-based RL (PbRL) has gained considerable interest, as comparative feedback is easy for humans to provide yet informative enough to guide agents (Kaufmann et al., 2024; Casper et al., 2023). By querying human preferences over pairs of video clips, robot agents trained with PbRL have demonstrated the ability to perform novel behaviors (Christiano et al., 2017) and avoid reward exploitation (Lee et al., 2021a). With these promising results, PbRL has gained popularity in both online (Lee et al., 2021b; Cheng et al., 2024) and offline (Shin et al., 2023; Choi et al., 2024) settings. The PbRL framework often consists of two stages: reward learning from preferences, followed by policy optimization with the learned reward.

While PbRL methods can align agents with human intent, effective reward functions require adequate coverage of both state and action spaces to achieve strong downstream performance (Ibarz et al., 2018; Hejna & Sadigh, 2023). Consequently, reward learning in PbRL is costly, often requiring thousands of human queries (Christiano et al., 2017; Shin et al., 2023; Yuan et al., 2024). To mitigate this challenge, prior work has explored several approaches, including semi-supervised learning (Park et al., 2022; Marta et al., 2024), meta-learning (Hejna III & Sadigh, 2023), active learning (Wang et al., 2022a), and preference ranking (Hwang et al., 2023; Choi et al., 2024). Yet a fundamental aspect remains underexplored—human preferences are shaped by the visual perception of agent behaviors, and leveraging these perceptual distinctions offers a promising direction for improving feedback efficiency. Our key insight is that the expressive and structured representation space of Video Foundation Models (ViFMs)–pretrained on large-scale video corpora–can be harnessed to infer preferences for new behaviors by

comparing them with known preferred examples.

To that end, we introduce Video-based Optimal Transport Preference labeling (VOTP), an algorithm that uses optimal transport over the ViFM representation space to automatically assign preference labels to unlabeled segment pairs given only a small number of labeled preference queries (*e.g.*, 10 feedbacks). Notably, unlabeled segment pairs can be obtained at no additional cost in PbRL settings, *e.g.*, from offline datasets. These pseudo-labeled segment pairs, together with the labeled ones, are then used to learn a reward function. Specifically, VOTP uses optimal transport to compute alignments between labeled and unlabeled pairs in the ViFM latent space. The pseudo-label for an unlabeled pair is then inferred by aggregating preferences from labeled pairs, weighted by their relative alignments computed from the optimal transport plan. Extensive experiments across simulated domains, including D4RL locomotion (Fu et al., 2020) and MetaWorld manipulation (Yu et al., 2020), as well as real-world tabletop manipulation tasks demonstrate that VOTP learns effective policies from limited preference labels, substantially improving feedback efficiency in PbRL. We further present extensive analyses and ablation studies to provide insights into VOTP's performance gains.

## 2. Related Work

**Preference-based RL (PbRL).**   PbRL enables agents to align with human intent through pairwise comparisons of behaviors, removing the need for manual reward engineering (Christiano et al., 2017). However, its scalability is constrained by the large amount of costly and labor-intensive human feedback it requires. To improve feedback efficiency, prior work has explored several directions, such as informative query selection (Bıyık et al., 2020; Wang et al., 2022a; Mu et al., 2025), pre-training of RL agents (Ibarz et al., 2018; Lee et al., 2021a), exploration guided by reward uncertainty (Liang et al., 2022), and preference rankings (Hwang et al., 2023; Choi et al., 2024). Other methods leverage pre-collected (sub-optimal) data to pre-train reward functions (Hejna III & Sadigh, 2023; Muslimani & Taylor, 2025). In contrast, we utilize unlabeled segment pairs from offline datasets for reward learning. Unlike (Park et al., 2022), which depends on learned reward models to perform pseudo-labeling, we employ optimal transport within the semantically meaningful latent space of Video Foundation Models (ViFMs) to infer pseudo-labels. This enables VOTP to learn effective reward functions from only a handful of preference feedbacks.

**Vision Foundation Models in Reward Learning.**   With the rapid progress of foundation models, recent studies have explored their potential in constructing reward functions. One line of work leverages pretrained vision-language mod-els (VLMs) to directly reward RL agents by measuring alignments between trajectories and task descriptions (Cui et al., 2022; Rocamonde et al., 2024; Sontakke et al., 2024; Lee et al., 2025). However, these reward signals are often noisy and inconsistent (Wang et al., 2024). Another line of research utilizes the reasoning ability of VLMs to provide feedback (Wang et al., 2024; Luu et al., 2025a; Venkataraman et al., 2025; Luu et al., 2025b). Yet such approaches rely on carefully crafted prompt templates to be effective. In this work, we instead leverage ViFMs to generate pseudo-preference labels, aiming to enhance the feedback efficiency of PbRL.

**Optimal Transport in Reinforcement Learning.**   Optimal Transport (OT) (Cuturi, 2013; Peyré & Cuturi, 2019) has been widely studied in domain adaptation (Courty et al., 2016), graph matching (Titouan et al., 2019; Ratnayaka et al., 2025), and semi-supervised learning (Tai et al., 2021; Tan et al., 2024). In the context of RL, prior works have applied OT to imitation learning (Fickinger et al., 2022; Luo et al., 2023; Fu et al., 2024; Huey et al., 2025) by minimizing the Wasserstein distance between the learner's trajectories and expert demonstrations. PEARL (Liu et al., 2024) extended this idea to transfer preferences across domains, but its applicability is restricted to tasks with identical state and action spaces, and cross-domain transfer often introduces high uncertainty for the target task. In contrast, VOTP performs pseudo-labeling directly within the same domain and scales naturally to high-dimensional visual inputs, enabling more stable and reliable reward learning in scenarios where PEARL is not applicable.

## 3. Preliminaries

**Reinforcement Learning.**   In reinforcement learning (RL), an agent interacts with an environment modeled as a Markov decision process (MDP). An MDP is defined by the tuple $\langle \mathcal{S}, \mathcal{A}, \mathcal{T}, r, \gamma \rangle$. At each time step $t$, the agent receives a state $\mathbf{s}_t \in \mathcal{S}$ and selects an action $\mathbf{a}_t \in \mathcal{A}$ based on its policy $\pi$. The environment responds by emitting a reward $r_t$ and transitioning to the next state $\mathbf{s}_{t+1}$ according to the transition probability $\mathcal{T}(\mathbf{s}'|\mathbf{s}, \mathbf{a})$. In our setting, we also consider the observation $\mathbf{o}_t \in \mathcal{O}$, which is an image rendered from the underlying state $\mathbf{s}_t$. The return, $G_t = \sum_{k=0}^{\infty} \gamma^k r(\mathbf{s}_{t+k}, \mathbf{a}_{t+k})$, is defined as the discounted cumulative sum of rewards, with discount factor $\gamma \in [0, 1)$. The objective of RL algorithms is to learn a policy that maximizes the expected return.

**Preference-based RL.**   In offline preference learning, we assume that the true reward function is unknown and instead learn a reward function $\widehat{r}_\psi$ from human preferences (Christiano et al., 2017; Ibarz et al., 2018). A trajectory segment of length $H$ is represented as a sequence of

states and actions $\{(\mathbf{s}_1, \mathbf{a}_1), \ldots, (\mathbf{s}_H, \mathbf{a}_H)\}$. Given a pair of segments $(\sigma^0, \sigma^1)$, a teacher provides a preference label $\tilde{y} \in \{0, 1, 0.5\}$, where $\tilde{y} = 0$ indicates $\sigma^0 \succ \sigma^1$, $\tilde{y} = 1$ indicates $\sigma^1 \succ \sigma^0$, and $\tilde{y} = 0.5$ indicates equal preference. Here, $\sigma^i \succ \sigma^j$ denotes that segment $i$ is preferred over segment $j$. Each feedback is stored in a preference dataset $\mathcal{D}$ as a triple $(\sigma^0, \sigma^1, \tilde{y})$. The preference predictor is modeled using the reward function $\widehat{r}_\psi$ following the Bradley-Terry model (Bradley & Terry, 1952):

$$P(\sigma^0 \succ \sigma^1; \psi) = \frac{\exp \sum_t \widehat{r}_\psi(\mathbf{s}_t^0, \mathbf{a}_t^0)}{\exp \sum_t \widehat{r}_\psi(\mathbf{s}_t^0, \mathbf{a}_t^0) + \exp \sum_t \widehat{r}_\psi(\mathbf{s}_t^1, \mathbf{a}_t^1)}.$$

Given the preference dataset, the estimated reward function $\widehat{r}_\psi$ is updated by minimizing the cross-entropy loss between predicted preferences and annotated labels:

$$\mathcal{L}(\psi) = \mathbb{E}_\mathcal{D}[(1 - \tilde{y}) \log P(\sigma^0 \succ \sigma^1; \psi) + \tilde{y} \log P(\sigma^1 \succ \sigma^0; \psi)]. \tag{1}$$

In practice, a preference query is typically presented to teachers as a pair of short video clips rendered from trajectory segments. While intuitive, learning an effective reward model often demands hundreds to thousands of annotated comparisons (Kim et al., 2023; Hejna & Sadigh, 2023; Hejna et al., 2024; Choi et al., 2024), which creates an unsustainable annotation burden. To mitigate this challenge, we adopt the semi-supervised preference learning paradigm (Park et al., 2022), which leverages both labeled and unlabeled segment pairs for reward learning.

**Discrete Optimal Transport.** Optimal Transport (OT) (Cuturi, 2013; Peyré & Cuturi, 2019) is an optimization problem that finds a coupling between two probability measures with minimal cost. Let $\Delta_n = \{\mathbf{p} \in \mathbb{R}_+^n \mid \sum_{i=1}^n p_i = 1\}$ denote the probability simplex of dimension $n$. Consider two probability measures $\mu_x = \sum_{i=1}^n p_i \delta_{x_i}$ and $\mu_y = \sum_{j=1}^m q_j \delta_{y_j}$, supported on $\{x_i\}_{i=1}^n$ and $\{y_j\}_{j=1}^m$, respectively. Here, the weight vector $\mathbf{p} = (p_1, \ldots, p_n)$ and $\mathbf{q} = (q_1, \ldots, q_m)$ belong to $\Delta_n$ and $\Delta_m$, respectively, and $\delta_x$ denotes the Dirac measure at $x$. The discrete OT problem between $\mu_x$ and $\mu_y$ can then be expressed via the Wasserstein distance as:

$$\mathcal{W}_2^2(\mu_x, \mu_y) = \min_{\mu \in \mathcal{M}} \sum_{i=1}^n \sum_{j=1}^m c(x_i, y_j) \mu_{ij}, \tag{2}$$

where $\mathcal{M} = \{\mu \in \mathbb{R}_+^{n \times m} : \mu \mathbf{1}_m = \mu_x, \mu^\top \mathbf{1}_n = \mu_y\}$ is the set of feasible transport plans, $\mathbf{1}_n$ denotes the all-ones vector of dimension $n$, and $c(x, y)$ is the cost function. The matrix $\mu$ specifies a transport plan, where $\mu_{ij}$ indicates the mass moved from $x_i$ to $y_j$. In this work, we leverage OT to compute correspondences between unlabeled and labeled segment pairs, thereby enabling the inference of pseudo-preference labels.

## 4. Method

Our goal is to improve feedback efficiency in offline preference learning by leveraging unlabeled data. To this end, we introduce Video-based Optimal Transport Preference (VOTP), a semi-supervised framework that infers pseudo-preference labels using optimal transport (OT). The framework consists of two key components: (i) trajectory representation with Video Foundation Models, and (ii) pseudo-preference label generation through the optimal transport plan. An overview is provided in Figure 1.

### 4.1. Trajectory Representation

Representing trajectory segments in a form that enables reliable comparison is central to preference learning (Tian et al., 2024; Mu et al., 2025). We model each segment as a short video clip, $\sigma = \{\mathbf{o}_1, \ldots, \mathbf{o}_H\}$, and embed it into a latent space using a trajectory encoder $f_\phi$:

$$\mathbf{z} = f_\phi(\mathbf{o}_{1:H}). \tag{3}$$

An effective encoder must capture both spatial details within frames and temporal dynamics across the segment, as these jointly determine the behavioral differences reflected in human preferences. To meet these requirements, we adopt off-the-shelf video foundation models (ViFMs) (Madan et al., 2024), which are pretrained on massive collections of human activity videos covering diverse actors, viewpoints, lighting conditions, and backgrounds. This large-scale, heterogeneous pre-training produces actor-agnostic, semantically rich embeddings that are robust to nuisance variation and generalize to unseen robotic environments.

### 4.2. Pseudo-Preference Label Generation

VOTP first identifies correspondences between labeled and unlabeled segment representations, and then assigns preferences via an OT plan. We denote the labeled dataset as $\mathcal{D}_l = \{(\sigma^0, \sigma^1, \tilde{y})^{(i)}\}_{i=1}^{N_l}$ and the unlabeled dataset as $\mathcal{D}_u = \{(\bar{\sigma}^0, \bar{\sigma}^1)^{(i)}\}_{i=1}^{N_u}$. Our objective is to infer pseudo labels for $\mathcal{D}_u$ and use both datasets to learn the reward function $\widehat{r}_\psi$.

We define the labeled set as $L = \{\sigma_i\}_{i=1}^N$, where $N = 2N_l$ denotes the total number of segments in $\mathcal{D}_l$. Preference relations among segments are encoded in a preference matrix $R \in \{-1, 0, 1\}^{N \times N}$:

$$R_{ij} = \begin{cases} -1 & \text{if } \sigma_i \succ \sigma_j, \\ 1 & \text{if } \sigma_j \succ \sigma_i, \\ 0 & \text{for } i = j, \text{ ties, or no preference is available.} \end{cases}$$

By construction, $R$ is skew-symmetric, i.e., $R^\top = -R$. In parallel, we define the unlabeled set $U = \{\bar{\sigma}_{i'}\}_{i'=1}^M$, consisting of $M$ segments sampled from $\mathcal{D}_u$, for which pseudo-preference labels are inferred. Let $\mu_L = \sum_{i=1}^N p_i \delta_{\sigma_i}$ and $\mu_U = \sum_{i'=1}^M q_{i'} \delta_{\bar{\sigma}_{i'}}$ denote the empirical measures on these

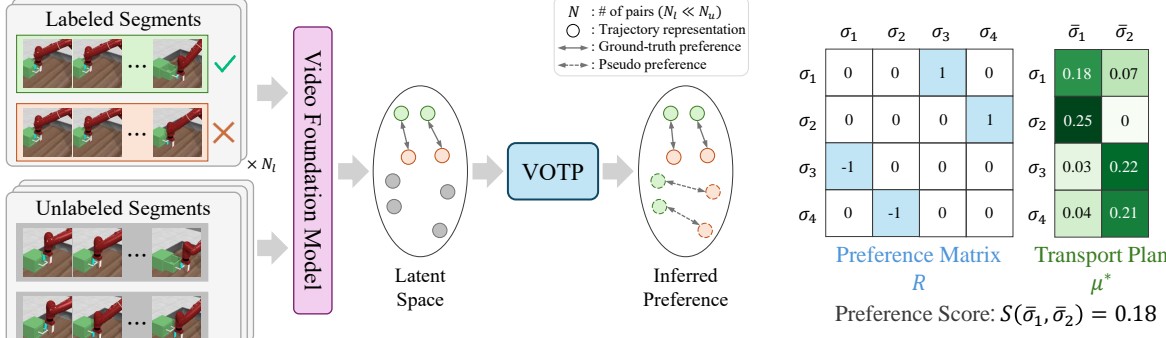

*(a)* Pipeline of Video-based Optimal Transport Preference (VOTP)  *(b)* Computation performed by VOTP

*Figure 1.* Overview of our framework. (a) VOTP embeds visual segments into a latent space using an off-the-shelf video foundation model and uses the optimal transport plan to propagate preferences with relative alignment strengths. Green dots indicate preferred segments over orange ones. (b) Example computation in VOTP with four labeled segments ($\sigma_i$) and two unlabeled segments ($\bar{\sigma}_{i'}$). Preference relations among labeled segments are represented by the preference matrix $R$. Each entry of the optimal transport plan $\mu^*$ specifies the probability that a labeled segment matches an unlabeled segment, and the unnormalized preference score is computed using Eq. (5).

sets. For simplicity, we adopt the uniform weights, *i.e.*, $p_i = \frac{1}{N}$ and $q_{i'} = \frac{1}{M}$. The OT plan for aligning labeled and unlabeled segments is then obtained as:

$$\mu^* = \arg\min_{\mu \in \mathcal{M}} \sum_{i=1}^{N} \sum_{i'=1}^{M} c(\sigma_i, \bar{\sigma}_{i'}) \mu_{ii'}, \quad (4)$$

where $\mathcal{M} = \{\mu \in \mathbb{R}_+^{N \times M} : \mu \mathbf{1}_M = \frac{1}{N}\mathbf{1}_N, \mu^\top \mathbf{1}_N = \frac{1}{M}\mathbf{1}_M\}$. The cost function is defined as $c(\sigma_i, \bar{\sigma}_{i'}) = d(f_\phi(\sigma_i), f_\phi(\bar{\sigma}_{i'}))$, the distance between encoded visual segments in the latent video space, where $d$ can be chosen as either the Euclidean distance or the cosine distance.

The OT plan $\mu^*$ obtained in Eq. (4) provides the correspondences between segments in sets $L$ and $U$. Concretely, each entry $\mu_{ii'}$ represents the probability that the unlabeled segment $\bar{\sigma}_{i'}$ matches the labeled segment $\sigma_i$. Combining these probabilities with the preference matrix $R$, we can infer preferences between segments in the unlabeled set $U$. For brevity, we denote the OT plan as $\mu$. We then define the preference score used to determine the preference between the unlabeled pair $(\bar{\sigma}_{i'}, \bar{\sigma}_{j'})$ as follows:

$$S(\bar{\sigma}_{i'}, \bar{\sigma}_{j'}) = \sum_{i=1}^{N} \sum_{j=1}^{N} R_{ij}(\mu_{ii'}\mu_{jj'} - \mu_{ij'}\mu_{ji'}) \quad (5)$$

**Interpretation.** Consider a labeled pair $(i, j)$ with a non-zero preference (*i.e.*, $R_{ij} \neq 0$). Suppose $R_{ij} = 1$ (*i.e.*, $\sigma_j \succ \sigma_i$). The term $\mu_{ii'}\mu_{jj'}$ measures alignment between $(\sigma_i, \sigma_j)$ and $(\bar{\sigma}_{i'}, \bar{\sigma}_{j'})$, while $\mu_{ij'}\mu_{ji'}$ measures the alignment with the reversed pair $(\bar{\sigma}_{j'}, \bar{\sigma}_{i'})$. If the difference $(\mu_{ii'}\mu_{jj'} - \mu_{ij'}\mu_{ji'})$ is positive, then $(\bar{\sigma}_{i'}, \bar{\sigma}_{j'})$ likely shares the preference of $(\sigma_i, \sigma_j)$, implying $\bar{\sigma}_{j'} \succ \bar{\sigma}_{i'}$. Conversely, if the difference is negative, the preference is flipped, *i.e.*, $\bar{\sigma}_{i'} \succ \bar{\sigma}_{j'}$. The preference score for $(\bar{\sigma}_{i'}, \bar{\sigma}_{j'})$ is then obtained by aggregating alignment comparisons across all labeled pairs. Since $R$ is skew-symmetric, the inferred preference for $(\bar{\sigma}_{i'}, \bar{\sigma}_{j'})$ is consistent under swapping $(i, j)$. Examples of this computation are illustrated in Figure 1(b)

and in the Appendix. Overall, VOTP leverages the transport plan to propagate preferences from labeled to unlabeled pairs through relative alignment strengths.

In practice, the entries of the OT plan $\mu$ are small because $\sum \mu_{ij} = 1$, which leads to relatively small preference scores. Therefore, we normalize the preference score by

$$S_{\max} = \sum_{i=1}^{N} \sum_{j=1}^{N} \frac{1}{N^2} \mathbb{1}(R_{ij} \neq 0). \quad (6)$$

Here, $S_{\max}$ denotes the absolute maximum attainable score under uniform masses (*i.e.*, $p_i = \frac{1}{N}$), assuming the OT plan maximizes the contribution of all non-zero $R_{ij}$ terms. This guarantees that preference scores lie within $[-1, 1]$ across varying numbers of labeled pairs. Finally, to obtain the preference label for the pair $(\bar{\sigma}_{i'}, \bar{\sigma}_{j'})$, we apply a preference threshold $\tau_P$ to determine the label[1]:

$$\tilde{y} = \begin{cases} \frac{1}{2}(1 + \text{sign}(S_{\text{norm}}(\bar{\sigma}_{i'}, \bar{\sigma}_{j'}))) & \text{if } |S_{\text{norm}}| \geq \tau_P, \\ 0.5 & \text{otherwise.} \end{cases} \quad (7)$$

where $\text{sign}(x) = -1$ if $x < 0$, 1 if $x > 0$, and 0 if $x = 0$; and $S_{\text{norm}} = S(\bar{\sigma}_{i'}, \bar{\sigma}_{j'})/S_{\max}$.

### 4.3. Implementation Details

Obtaining the optimal coupling matrix $\mu^*$ in Eq. (4) requires solving a linear program, which is computationally expensive with standard solvers. In practice, we solve the entropy-regularized OT problem using Sinkhorn's algorithm (Cuturi, 2013), which provides both efficiency and numerical stability. Our implementation uses the Sinkhorn solver from the POT toolbox (Flamary et al., 2021). After VOTP annotates the unlabeled dataset with pseudo-preferences, we train the reward function $\hat{r}_\psi$ using Eq. (1). To mitigate the impact of inaccurate pseudo-labels, we retain only those

---

[1]One could optionally apply an additional threshold to treat pairs with scores near zero as equally preferable.

with scores above the threshold $\tau_P$. During RL training, all state-action pairs in the offline dataset are relabeled using the trained $\widehat{r}_\psi$. The overall procedure is summarized in Algorithm 1 in the Appendix.

## 5. Experiments

In this section, we conduct experiments across diverse domains to answer the following questions: (1) Can VOTP improve feedback efficiency in a low-data regime? (2) What is the individual contribution of each component within VOTP? (3) How do key hyperparameters influence the performance of VOTP? (4) How does VOTP generalize to unseen environments with visual distractors? (5) Can VOTP be effectively applied to real robotic tasks?

### 5.1. Setups

**Benchmark.** To conduct a comprehensive evaluation of our method, we select several continuous control tasks from two benchmarks, D4RL (Fu et al., 2020) and MetaWorld (Yu et al., 2020), where D4RL provides locomotion tasks and MetaWorld involves manipulation tasks. We use offline PbRL datasets from Kim et al. (2023) for D4RL and Hejna et al. (2024) for MetaWorld. We initialize the labeled set with 10 segment pairs and use synthetic preferences generated by scripted teachers based on the ground-truth reward function[2], following common practice in PbRL evaluation (Lee et al., 2021b; Shin et al., 2023; Choi et al., 2024). For pseudo-labeling, we sample additional pairs uniformly at random from the offline datasets. Specifically, we use a total of 10k queries for D4RL and 50k queries for MetaWorld. Since VOTP performs labeling on image-based observations, we additionally render visual observations corresponding to the states in the preference datasets.

**Training and Evaluation.** For computing the optimal coupling, we use the Sinkhorn solver from POT (Flamary et al., 2021), a library for optimal transport that provides efficient computation of the Sinkhorn algorithm with accelerator support. We use the Euclidean distance as the cost function. As the trajectory encoder, we adopt S3D (Xie et al., 2018; Miech et al., 2020), a ViFM pretrained on HowTo100M (Miech et al., 2019), which consists of large-scale third-person clips of everyday human activities. For reward learning, we use both labeled and pseudo-labeled pairs, retaining pseudo-labels above the threshold $\tau_P$ (Eq. 7). After training the reward model, we replace the original rewards in the offline dataset with the learned rewards and then train the policy using an offline RL algorithm. VOTP can be applied to any offline RL algorithm, but as in prior

---

[2]For hopper-medium-replay-v2, we use human labels from Kim et al. (2023), since scripted labels remain ineffective across baselines even when provided in large quantities.

work, we use IQL (Kostrikov et al., 2022). Across PbRL baselines, both the policy and reward models are trained from states and share the same policy-learning hyperparameters. Thus, the only difference lies in the reward learning process. Further implementation details can be found in the Appendix. We evaluate performance using normalized scores on D4RL and success rates on MetaWorld. For all experiments, we report the mean and standard deviation across five runs, with each run evaluated with 25 episodes per evaluation step. Full learning curves and interquartile mean (IQM) (Agarwal et al., 2021) results are provided in the Appendix.

### 5.2. Evaluation on Offline PbRL Benchmark

**Baselines.** We compare VOTP with a diverse range of offline PbRL baselines. These include methods *without explicit reward modeling*: Inverse Preference Learning (IPL) (Hejna & Sadigh, 2023), Contrastive Preference Learning (CPL) (Hejna et al., 2024), and Direct Preference-based Policy Optimization (DPPO) (An et al., 2023); and those *with explicit reward modeling*: a preference-based variant of IQL (P-IQL) (Hejna et al., 2024), Semi-supervised reward learning (SURF) (Park et al., 2022), Listwise Reward Estimation (LiRE) (Choi et al., 2024), Adversarial Preference-based Policy Optimization (APPO) (Kang & Oh, 2025), and Flow-to-Better (FTB) (Zhang et al., 2024). Among the latter, P-IQL refers to the method trained on only the labeled dataset. LiRE builds a ranked list of feedback to exploit second-order information, and APPO frames PbRL as a two-player game between a policy and a dynamics model. Finally, both SURF and FTB utilize unlabeled datasets to augment the preference dataset, similar to our approach: SURF relies on trained reward models to generate pseudo-labels, while FTB relies on a diffusion model to generate better-preferred trajectories. Additionally, we add an Oracle baseline, which is P-IQL trained similarly to ours, but using synthetic preferences for the unlabeled dataset.

Table 1 summarizes the performance of offline RL using ground-truth (GT) rewards versus preference feedback. We find that methods without explicit reward modeling perform poorly compared to GT reward training, while methods that learn a reward model generally achieve better results. This suggests that explicit reward modeling is more robust under low-data regimes, as also observed in Zhang et al. (2024). Among reward modeling methods, SURF improves upon P-IQL in MetaWorld tasks and *walker2d*, yet its performance degrades in *hop-\**. This is likely due to inaccurate pseudo-labels generated by reward models under limited supervision, leading to confirmation bias (Chen et al., 2022a; Tan et al., 2024). While APPO shows gains in MetaWorld, it appears detrimental in D4RL tasks. FTB likewise utilizes few labels and generally outperforms P-IQL in both domains. However, a significant drawback of FTB is its computa-

*Table 1.* Average performance of methods on D4RL locomotion and MetaWorld. For D4RL tasks, "hop" denotes hopper, while "m", "r", and "e" denote medium, replay, and expert, respectively. We run five seeds and report the final performance at the end of training like Kostrikov et al. (2022). Bold values indicate results within 5% of the best-performing method (excluding IQL+GT and Oracle).

| Dataset | IQL+GT | Oracle | IPL | CPL | DPPO | P-IQL | SURF | LiRE | APPO | FTB | VOTP |
|---|---|---|---|---|---|---|---|---|---|---|---|
| hop-m-r | $87.5_{\pm7.4}$ | $91.3_{\pm3.2}$ | $22.1_{\pm4.9}$ | $6.5_{\pm0.5}$ | $59.4_{\pm12.0}$ | $36.5_{\pm15.4}$ | $9.3_{\pm0.6}$ | $52.1_{\pm26.9}$ | $1.3_{\pm0.6}$ | $\mathbf{90.5}_{\pm3.9}$ | $\mathbf{91.1}_{\pm4.7}$ |
| hop-m-e | $104.5_{\pm4.5}$ | $101.9_{\pm4.5}$ | $62.6_{\pm18.4}$ | $54.2_{\pm7.0}$ | $70.1_{\pm20.2}$ | $89.1_{\pm18.4}$ | $65.5_{\pm17.0}$ | $106.3_{\pm7.2}$ | $40.7_{\pm16.7}$ | $\mathbf{111.9}_{\pm0.7}$ | $105.7_{\pm6.0}$ |
| walker2d-m-r | $72.6_{\pm4.9}$ | $66.9_{\pm10.3}$ | $8.6_{\pm5.4}$ | $6.6_{\pm0.6}$ | $35.1_{\pm6.9}$ | $32.4_{\pm27.1}$ | $64.9_{\pm9.4}$ | $\mathbf{71.3}_{\pm16.0}$ | $12.8_{\pm8.1}$ | $62.8_{\pm8.8}$ | $66.3_{\pm5.6}$ |
| walker2d-m-e | $109.9_{\pm0.5}$ | $109.6_{\pm0.8}$ | $92.4_{\pm10.2}$ | $39.9_{\pm19.9}$ | $70.2_{\pm12.4}$ | $103.4_{\pm7.0}$ | $103.2_{\pm15.1}$ | $\mathbf{109.7}_{\pm1.1}$ | $31.2_{\pm6.2}$ | $76.5_{\pm2.2}$ | $\mathbf{108.1}_{\pm2.2}$ |
| loco avg. | 93.6 | 92.4 | 46.4 | 26.8 | 58.7 | 65.3 | 59.5 | 83.2 | 21.5 | 85.4 | **92.8** |
| door-open | $79.2_{\pm5.9}$ | $90.4_{\pm3.2}$ | $48.8_{\pm11.7}$ | $44.8_{\pm22.3}$ | $14.4_{\pm12.1}$ | $36.8_{\pm13.2}$ | $74.4_{\pm10.3}$ | $\mathbf{84.0}_{\pm4.9}$ | $73.6_{\pm7.8}$ | $43.2_{\pm6.6}$ | $\mathbf{84.0}_{\pm8.4}$ |
| drawer-open | $83.2_{\pm4.7}$ | $80.4_{\pm7.4}$ | $51.2_{\pm13.5}$ | $52.8_{\pm13.9}$ | $59.0_{\pm12.4}$ | $36.0_{\pm13.6}$ | $57.6_{\pm15.7}$ | $70.0_{\pm6.3}$ | $64.8_{\pm14.8}$ | $52.8_{\pm7.2}$ | $\mathbf{71.2}_{\pm11.7}$ |
| plate-slide | $56.0_{\pm11.9}$ | $62.4_{\pm4.8}$ | $28.0_{\pm9.1}$ | $34.4_{\pm12.0}$ | $37.1_{\pm17.1}$ | $15.2_{\pm5.9}$ | $23.2_{\pm5.9}$ | $38.0_{\pm7.5}$ | $24.0_{\pm17.2}$ | $41.6_{\pm3.6}$ | $\mathbf{57.6}_{\pm5.4}$ |
| sweep-into | $65.6_{\pm5.4}$ | $59.0_{\pm14.2}$ | $41.6_{\pm3.2}$ | $37.6_{\pm6.0}$ | $48.0_{\pm14.8}$ | $36.0_{\pm8.0}$ | $40.8_{\pm4.7}$ | $\mathbf{64.0}_{\pm10.2}$ | $44.0_{\pm16.2}$ | $56.8_{\pm13.7}$ | $57.6_{\pm7.4}$ |
| mw avg. | 71.0 | 80.1 | 42.4 | 42.4 | 39.6 | 31.0 | 51.0 | 64.0 | 51.6 | 48.6 | **67.6** |

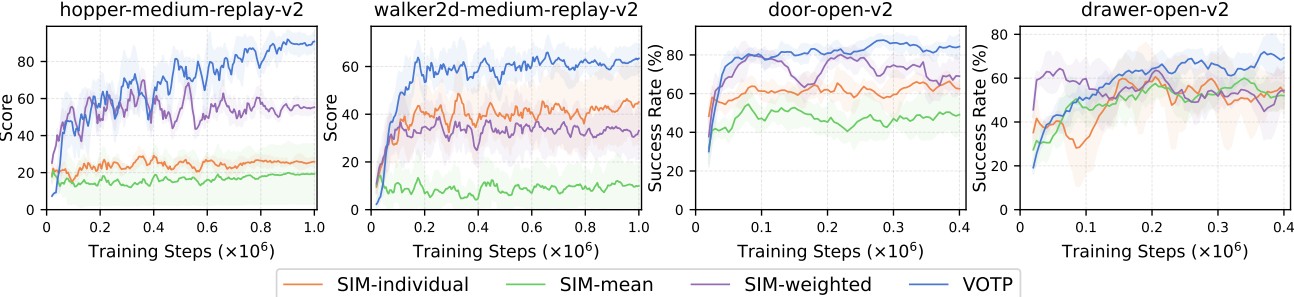

*Figure 2.* The effectiveness of using OT to infer pseudo-labels. Results are averaged over five runs with standard deviation (shaded area).

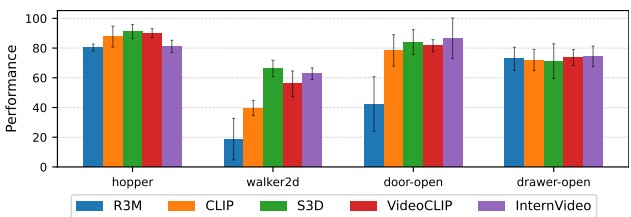

*Figure 3.* Ablation with various trajectory encoders in D4RL and MetaWorld. For *hopper* and *walker2d*, we use medium-replay datasets. Results are averaged over five runs.

tional overhead; training requires approximately 2 days per run, whereas VOTP achieves superior performance in less than 2 hours. Finally, while LiRE consistently improves over P-IQL, it remains significantly behind the IQL+GT and Oracle in tasks such as *hop-m-r* and *plate-slide*.

In contrast, VOTP achieves substantial performance gains over P-IQL across all domains and effectively matches the Oracle performance on D4RL. These results highlight the effectiveness of OT-based pseudo-labeling when combined with the rich latent space of ViFMs to generate high-quality pseudo-preferences. In addition, policies trained with preference-based rewards outperform those trained with GT rewards on several tasks, suggesting that reward models learned from preference data can be more effective, as also reported in prior works (Christiano et al., 2017; Kim et al., 2023; An et al., 2023; Choi et al., 2024). Finally, our framework is compatible with orthogonal techniques such as active query selection (Wang et al., 2022a; Mu et al., 2025) and preference ranking (Choi et al., 2024), potentially

offering further gains in feedback efficiency.

### 5.3. Ablation Studies

**Effect of Video Foundation Models.** We assess the role of the video encoder in VOTP by comparing image foundation models (IFMs) and video foundation models (ViFMs) in encoding visual segments. For IFMs, we adopt R3M (Nair et al., 2022) and CLIP (Radford et al., 2021), which are widely used for feature extraction and reward computation (Adeniji et al., 2023; Zhang et al., 2023; Rocamonde et al., 2024). For ViFMs, we adopt S3D (Xie et al., 2018; Miech et al., 2020), VideoCLIP (Xu et al., 2021), and InternVideo (Wang et al., 2022b). As shown in Figure 3, we observe that ViFMs generally perform better than IFMs, particularly in *walker2d* and *door-open*. This improvement highlights their advantage in providing richer segment representations by capturing temporal dynamics and subtle motion cues, which are crucial for distinguishing behavioral differences when determining preferences. In our framework, we opt for S3D because it achieves robust performance across tasks while requiring far fewer parameters (31M) than VideoCLIP (208M) and InternVideo (478M). We expect that performance could be further improved by leveraging more advanced ViFMs.

**Effect of Optimal Transport.** To assess the benefits of our proposed OT formulation in pseudo-label inference, we compare against baselines that infer preferences solely based on similarity. Specifically, we divide the labeled set into preferred and non-preferred groups. The first baseline,

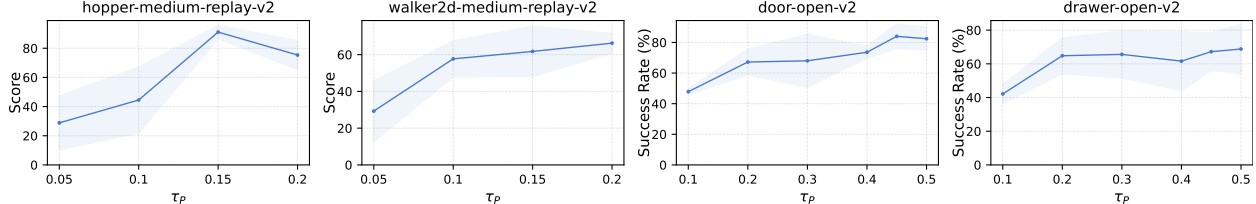

*Figure 4.* Average performance of each method as the number of preference feedbackts varies.

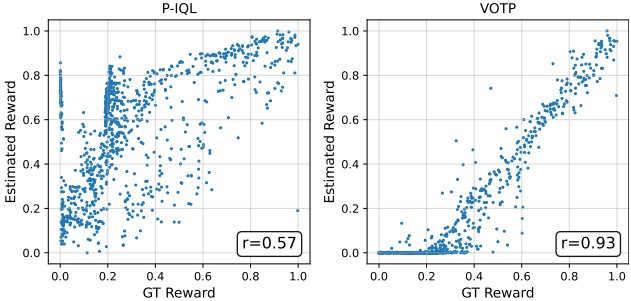

*Figure 5.* Performance of VOTP under different values of the preference threshold $\tau_P$.

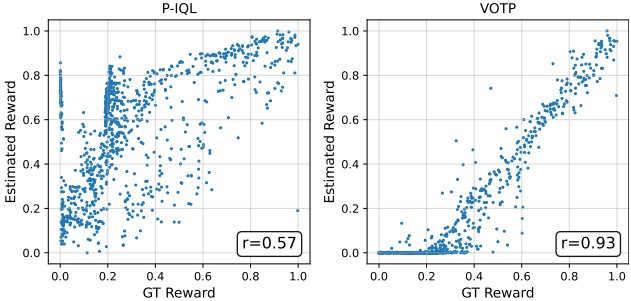

*Figure 6.* The Pearson correlation ($r$) between learned rewards and GT rewards for P-IQL and VOTP in *door-open*.

*SIM-individual*, assigns the label of the most similar labeled pair to an unlabeled pair. The second baseline, *SIM-mean*, instead compares with the aggregated representation of each group, obtained by averaging feature vectors. The third baseline, *SIM-weighted*, assigns labels based on a similarity-weighted average of preferences from the labeled pairs. In contrast, VOTP aggregates all preference labels from labeled pairs, weighting their contributions by the relative alignment strengths computed from the OT plan, thereby producing more reliable pseudo-labels. The results in Figure 2 demonstrate a clear advantage of our method. We also observe that *SIM-mean* performs worse than *SIM-individual*, likely because averaging group features discards fine-grained distinctions between pairs, which are crucial for assigning pseudo-preferences. Although *SIM-weighted* improves over *SIM-individual* on some tasks, its overall performance remains noticeably lower and less stable. This underscores the effectiveness of our OT for generating robust pseudo-labels.

**Varying the number of queries.** We evaluate how the number of queries affects PbRL performance in two domains: D4RL and MetaWorld. Concretely, we measure the

average performance of P-IQL, SURF, and VOTP while varying the labeled dataset size, ranging from 5 to 1000 preference labels depending on the domain. We note that most previous work on D4RL uses up to 500 preferences (Kim et al., 2023; An et al., 2023), while MetaWorld typically uses up to 10k (Hejna & Sadigh, 2023; Hejna et al., 2024). Results are shown in Figure 4. In D4RL, without pseudo-labels, P-IQL requires roughly 50-100 labels to match task-reward performance, whereas in MetaWorld it requires around 1k. Incorporating pseudo-labels improves performance in both domains. Importantly, we find that, except for *walker-medium-replay*, VOTP requires fewer labels than baselines to reach task-reward performance. Notably, in *door-open*, VOTP with only 10 labels outperforms the policy trained with ground-truth rewards. Overall, these results demonstrate the high feedback efficiency of VOTP, confirming its effectiveness in low-data regimes.

**Impact of the preference threshold.** We examine how the preference threshold $\tau_P$ affects the performance of VOTP. Concretely, we vary $\tau_P$ and measure the corresponding performance of VOTP. Results are shown in Figure 5. We observe that performance generally improves as the threshold increases, but slightly drops with a large value, as seen in *hopper-medium-replay*. This effect arises because our unlabeled dataset size is fixed due to the rendering cost of visual segments, and only pseudo-labels above $\tau_P$ are retained for training. Thus, increasing $\tau_P$ enhances label quality but reduces their quantity, which can harm performance. In practice, we tune this parameter to balance the quality-quantity trade-off of pseudo-labels by selecting values within the observed range of preference scores.

**Comparison of estimated rewards with GT rewards.** We

*Table 2.* Performance of VOTP under various types of visual distractions. Mean and standard deviation are computed over 5 random seeds.

| Dataset | Same Domain | Light (pos.+dir.) | Light (amb.+diff.) | Texture | Video (easy) | Video (hard) |
|---|---|---|---|---|---|---|
| door-open | $84.0_{\pm 8.4}$ | $88.8_{\pm 3.0}$ | $79.2_{\pm 3.0}$ | $76.8_{\pm 12.7}$ | $79.2_{\pm 6.9}$ | $80.4_{\pm 4.1}$ |
| drawer-open | $71.2_{\pm 11.7}$ | $74.4_{\pm 9.2}$ | $77.6_{\pm 7.4}$ | $72.0_{\pm 4.4}$ | $68.0_{\pm 5.7}$ | $68.8_{\pm 8.2}$ |
| Average | 77.6 | 81.6 | 78.4 | 74.4 | 73.6 | 74.6 |

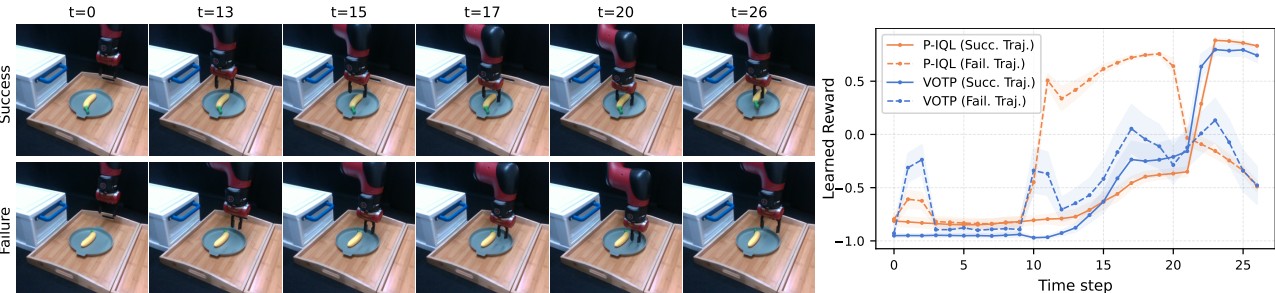

*Figure 7. Lift Banana*: Examples of successful and failed trajectories at each time step (left) with the corresponding reward outputs over timesteps from VOTP and P-IQL (right). Additional results and detailed experimental setup are provided in the Appendix.

examine the reward values estimated by the learned reward models of P-IQL and VOTP. Figure 6 presents scatter plots of estimated rewards against GT rewards in *door-open*. As illustrated, the reward estimates produced by VOTP exhibit a significantly stronger correlation with GT rewards compared to P-IQL. These results demonstrate that leveraging pseudo labels effectively improves coverage of both state and action spaces for reward learning, which in turn leads to a policy that performs much better. Additional results and analyses are provided in the Appendix.

### 5.4. Robustness to Nuisance Variation

One benefit of using ViFMs to encode visual segments is their strong generalization across varied visual conditions. To examine this ability, following Yuan et al. (2023), we modify MetaWorld environments with controlled visual distractions, including changes in lighting, visual appearance, and dynamic video backgrounds. Examples of these scenarios are shown in Figure 19 in the Appendix. To evaluate the robustness of VOTP, for each type of distraction we generate unlabeled visual segments under the corresponding perturbation while keeping the labeled segments fixed across scenarios. The results in Table 2 indicate that VOTP maintains strong performance across all settings and, in some cases, exhibits slight improvements under changes in lighting position and direction, suggesting robustness to a wide range of nuisance variations.

### 5.5. Real Robot Evaluation

We further evaluate VOTP in a real-world robotic manipulation setting using a 7-DoF Rethink Sawyer robotic arm. We compare our method against two baselines: Behavior Cloning (BC) and P-IQL. The experiments are conducted with two vision-based manipulation tasks: *Lift Banana* and

*Drawer Open*. In our setting, the policy input consists of proprioceptive states and image observations captured from a camera. For each task, we collect 50 demonstrations via keyboard teleoperation with a 50% success rate. To collect preferences, we present pairs of video clips to a human teacher. We use 5 and 10 preference labels for *Lift Banana* and *Drawer Open*, respectively. The number of unlabeled pairs is 2000 and 3000, respectively. The policy is trained using IQL (Kostrikov et al., 2022), with the reward model optimized according to Eq. (1). P-IQL and VOTP are trained in the same way as in the simulated experiments, *i.e.*, P-IQL is trained with a small number of labeled preferences, while VOTP is additionally trained with pseudo-labels. Table 3 reports the comparison with baselines, showing that by leveraging unlabeled data, VOTP enables the agent to achieve higher performance. To highlight the benefit of unlabeled data, Figure 7 shows reward outputs from VOTP and P-IQL on a successful and a failed trajectory. Both methods yield reasonable rewards for the successful trajectory, but P-IQL mistakenly assigns high rewards to failed behavior (timesteps 11-20). In contrast, VOTP produces well-separated rewards between successful and failed trajectories.

*Table 3.* Success rates over 10 episodes on real-world tasks.

| | Lift Banana | Drawer Open |
|---|---|---|
| BC | 20.0 | 40.0 |
| P-IQL | 50.0 | 50.0 |
| VOTP | **80.0** | **70.0** |

## 6. Discussion

In this work, we introduce Video-based Optimal Transport Preference (VOTP), a semi-supervised preference learning that employs optimal transport over latent space of ViFMs to automatically infer preferences for unlabeled pairs. This en-

ables VOTP to learn effective reward functions from only a handful of preference labels, substantially reducing the need for human supervision. Extensive experiments across locomotion, manipulation, and real-world robotic manipulation tasks validate the effectiveness of our approach, highlighting VOTP as a scalable and practical solution for PbRL.

**Limitations.** Since VOTP relies on pretrained ViFMs to generate pseudo-labels, any inherent biases in these models may be reflected in the learned reward function and, consequently, in the resulting policy. While this does not diminish the effectiveness of our approach, it suggests that careful evaluation of learned policies remains important before deployment in safety-critical applications. In addition, the computational cost of our method scales with the number of preference labels, as computing the OT plan becomes more expensive for larger datasets. While this cost remains manageable in our experiments, improving the scalability of the OT computation, for example via approximate or hierarchical transport (Halmos et al., 2025), represents an important direction for future work.

## Acknowledgements

This work was supported by Institute for Information & communications Technology Planning & Evaluation (IITP) grant funded by the Korea government(MSIT) (No.RS-2021-II211381, Development of Causal AI through Video Understanding and Reinforcement Learning, and Its Applications to Real Environments) and the National Research Foundation of Korea(NRF) grant funded by the Korea government(MSIT) (RS-2025-24742969, Intelligent Robotic System using Continual Learning and Multimodal Language Model based Multi Attribute Feedback).

## Impact Statement

This paper presents work whose goal is to advance the field of Machine Learning. There are many potential societal consequences of our work, none of which we feel must be specifically highlighted here.

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

# Appendix

---

**Algorithm 1** Pseudo-code for Video-based Optimal Transport Preference (VOTP)

---

1: **Input**: Offline dataset $\mathcal{B}$, labeled dataset $\mathcal{D}_l$, number of unlabeled segments $M$, threshold $\tau_P$.
2: **Initialize**: pseudo-labeled dataset $\mathcal{D}_u \leftarrow \emptyset$.
3: **for** each iteration **do**
4:     Sample $\frac{M}{2}$ segment pairs from $\mathcal{B}$
5:     Compute preference scores for segment pairs using Eq. (5)
6:     Assign pseudo-labels using Eq. (7) and append to $\mathcal{D}_u$
7: **end for**
8: Construct preference dataset $\mathcal{D} \leftarrow \mathcal{D}_l \cup \mathcal{D}_u$
9: Train reward model $\widehat{r}_\psi$ using Eq. (1)
10: Relabel rewards for state-action pairs in $\mathcal{B}$ using trained $\widehat{r}_\psi$
11: Train policy $\pi_\theta$ using an offline RL algorithm

---

# A. Details on Experiments

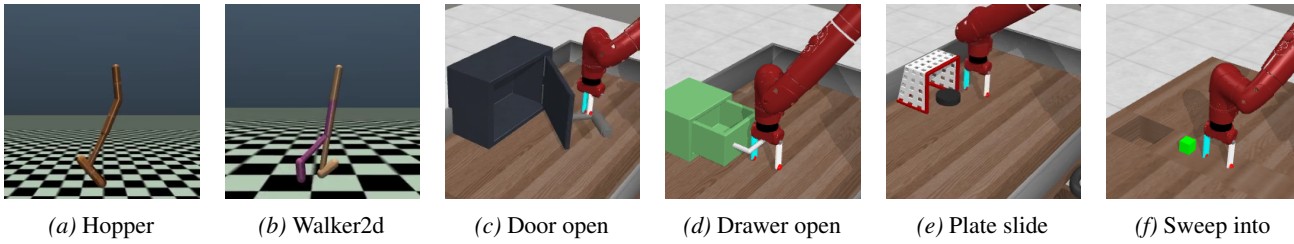

*(a)* Hopper      *(b)* Walker2d      *(c)* Door open      *(d)* Drawer open      *(e)* Plate slide      *(f)* Sweep into

*Figure 8.* Overview of environments used in our experiments: Gym Locomotion (a–b) and MetaWorld Manipulation (c–f).

## A.1. Task Details

The locomotion tasks from D4RL (Fu et al., 2020) and the manipulation tasks from MetaWorld (Yu et al., 2020) used in our experiments are shown in Figure 8.

**D4RL.** In D4RL locomotion tasks, the goal is to control simulated robots to move forward efficiently while minimizing energy costs for safe behavior. We use two tasks: Hopper and Walker2d, as in previous works (Kim et al., 2023; Hejna & Sadigh, 2023).

**MetaWorld.** In this domain, the agent produces low-level continuous actions to control a simulated 7-DoF Sawyer robotic arm, enabling interaction with tabletop objects to perform diverse manipulation tasks. Initial arm position is randomized. We evaluate four tasks: (1) *Door Open*: Open the door of a safe; (2) *Drawer Open*: Pull open a drawer; (3) *Plate Slide*: Slide a black plate into the designated goal region; and (4) *Sweep Into*: Sweep a green puck into the squared hole.

## A.2. Dataset Details

In offline preference-based RL, two types of data are provided: (i) an offline dataset collected from an unknown policy and (ii) a preference dataset consisting of pairs of trajectory segments sampled from the offline dataset. For D4RL locomotion, we use *medium-expert-v2*—which mixes equal portions of expert and partially trained demonstrations—and *medium-replay-v2*, which corresponds to the replay buffer of a partially trained policy. For MetaWorld, we use the pre-collected dataset from Hejna et al. (2024). For the preference dataset, we use pair indices from the publicly available datasets of Kim et al. (2023) and Hejna et al. (2024). For preference labels, we use scripted labels obtained from the ground-truth reward functions, except for *hopper-medium-replay-v2*, where we use human labels. Since the preference dataset from Kim et al. (2023) contains at most 500 preferences, we additionally generate pair indices for unlabeled data using the code from Kim et al. (2023).

## A.3. Implementation Details

The hyperparameters used in our main experiments are shown in Table 4, 5, and 6.

*Table 4.* Hyperparameters of IQL.

| Hyperparameter | D4RL | MetaWorld |
|---|---|---|
| Optimizer | Adam | Adam |
| Learning rate | 3e-4 | 3e-4 |
| Batch size | 256 | 512 |
| Hidden layer dim | 256 | 256 |
| Hidden layers | 2 | 2 |
| Activation | ReLU | ReLU |
| $\beta$ | 3.0 | 10.0 |
| $\tau$ | 0.7 | 0.9 |
| Training steps | 1e6 | 4e5 |

*Table 5.* Hyperparameters of the reward model.

| Hyperparameter | D4RL | MetaWorld |
|---|---|---|
| Optimizer | Adam | Adam |
| Learning rate | 3e-4 | 3e-4 |
| Batch size | 8 | 32 |
| Hidden layer dim | 256 | 128 |
| Hidden layers | 2 | 2 |
| Activation | ReLU | LeakyReLU |
| Output activation | Identity | Tanh |
| Segment length | 100 | 64 |
| Subsample length | 64 | 42 |
| Training steps | 2e4 | 2e4 |
| Score function (Choi et al., 2024) | Exponential | Linear |

*Table 6.* Hyperparameters of VOTP

| Hyperparameter | D4RL | MetaWorld |
|---|---|---|
| Total #labeled pairs | 10 | 10 |
| Total #unlabeled pairs | 10k | 50k |
| $M$ (in Alg. 1) | 2 | 2 |
| Distance metric in Eq. 5 | Euclidean | Euclidean |
| Preference threshold $\tau_P$ | $\{0.15, 0.2\}$ | $\{0.35, 0.4, 0.45\}$ |

## A.4. Real Robot Experiment Setups

We evaluate our method on two vision-based manipulation tasks using a 7-DoF Rethink Sawyer robotic arm in a tabletop environment. The tasks probe both reaching and object interaction and are defined as follows:

1. *Lift Banana*: grasp a banana from a plate and lift it.
2. *Drawer Open*: pull open a drawer beyond a fixed distance.

The robot is controlled with end-effector (EE) delta actions that command Cartesian displacements of the gripper. The EE orientation is constrained to yaw only, and control runs at 10Hz. For each task, the initial poses of a banana or a drawer handle are randomized within the workspace and observed by an Intel RealSense D435i RGB camera, which can be found in Figure 9). We collect 40 episodes for *lift banana* and 50 episodes for *drawer open* via keyboard teleoperation. Policies use both low-dimensional

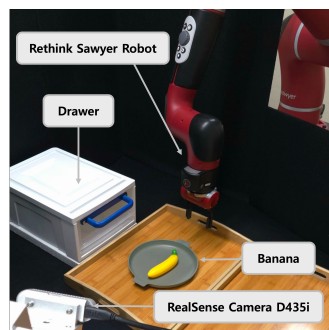

*Figure 9.* Setup in real robot.

states and visual observations. The visual observation is an RGB image at $480 \times 480$ resolution, resized to $224 \times 224$. We use ViFM to produce a 512-dimensional visual feature. The low-dimensional state is a 9-dimensional vector comprising the EE Cartesian position (3 dimensions), linear velocity (3 dimensions), yaw orientation (1 dimension), and the gripper status encoded as one-hot (open or closed, 2 dimensions). We concatenate the visual feature and the low-dimensional states to form a 521-dimensional input to the policy. All hyperparameter settings for the real-robot experiments can be found at Table 7. For evaluation, we measure success rate over 10 episodes per task.

## A.5. Offline PbRL Baselines

For baselines in Table 1, we use the official implementations provided in their publicly released repositories, as listed in Table 8. We obtain their performance by re-running the author-provided implementations under the same conditions as ours (*e.g.*, same number of labels) and conducting hyperparameter searches within the recommended ranges for a fair comparison. For FTB, we use the default hyperparameters, as training the method takes approximately two days. We report the average performance over 25 rollouts using the final checkpoint and run all experiments with 5 random seeds.

# B. Learning Curves

*Table 7.* Hyperparameters of real robot experiments.

|  | **Hyperparameter** | **Value** |
|---|---|---|
| IQL | Optimizer | Adam |
|  | Learning Rate | 3e-4 |
|  | Batch Size | 256 |
|  | $\beta/\tau$ | 3.0/0.7 |
|  | Training Steps | $1e5$ |
| Reward Model | Batch size | 8 |
|  | Activation | LeakyReLU |
|  | Output activation | Tanh |
|  | Segment length | 16 |
|  | Training steps | 2000 |
| VOTP | Total #labeled pairs | 5 (Lift), 10 (Drawer) |
|  | Total #unlabeled pairs | 2000 (Lift), 3000 (Drawer) |
|  | Preference threshold $\tau_P$ | 0.6 |

*Table 8.* Source code links and hyperparameter/variant search settings for all baselines.

| Algorithm | URL | Hyperparameters Tuning |
|---|---|---|
| IPL | https://github.com/jhejna/inverse-preference-learning | Default |
| CPL | https://github.com/jhejna/cpl | with/without BC |
| DPPO | https://github.com/snu-mllab/DPPO | $\lambda \in \{0.1, 0.5\}$, smooth $m \in \{5, 10, 15\}$ |
| SURL | https://github.com/alinlab/SURF | Threshold $\tau \in \{0.99, 0.999\}$ |
| LiRE | https://github.com/chwoong/LiRE | Budget $Q \in \{2, 4, 5, 10\}$ |
| APPO | https://github.com/oh-lab/APPO | $\lambda \in \{0.01, 0.3, 0.1\}$ |
| FTB | https://github.com/Zzl35/flow-to-better | Default |

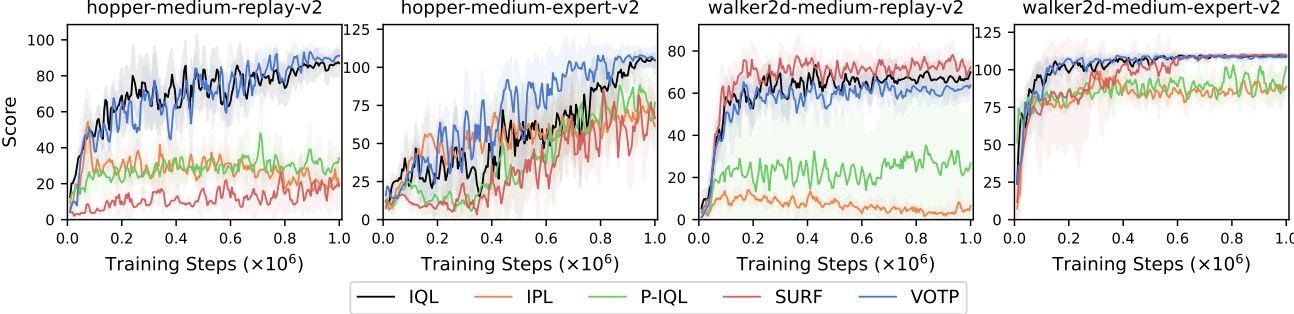

*Figure 10.* Learning curves of IQL, IPL, P-IQL, SURF, and VOTP on D4RL (Table 1). Results are means of 5 runs with standard deviation (shaded area). We smooth the learning curves using a moving average with a window size of 3.

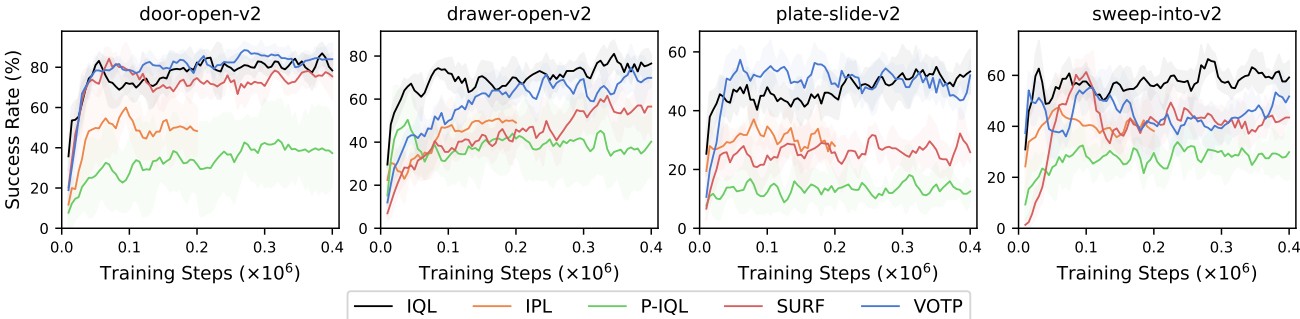

*Figure 11.* Learning curves of IQL, IPL, P-IQL, SURF, and VOTP on MetaWorld (Table 1). Results are means of 5 runs with standard deviation (shaded area). We smooth the learning curves using a moving average with a window size of 3.

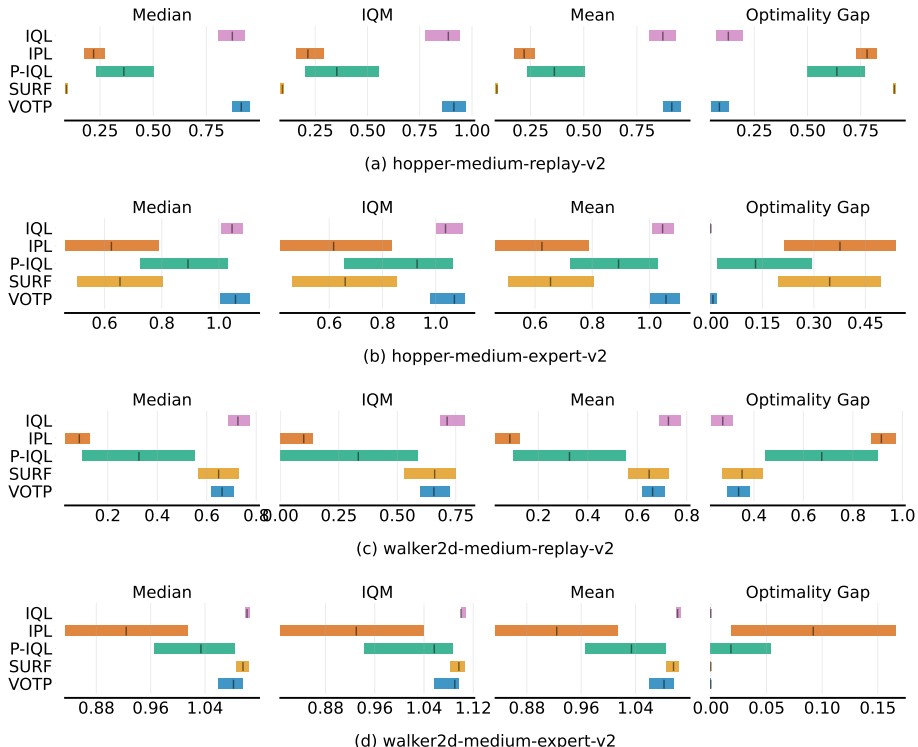

*Figure 12.* Aggregate metrics on D4RL locomotion tasks with 95% confidence intervals (CIs) across five runs. Higher mean, median and IQM scores and lower optimality gap are better. The CIs are estimated using the percentile bootstrap with stratified sampling.

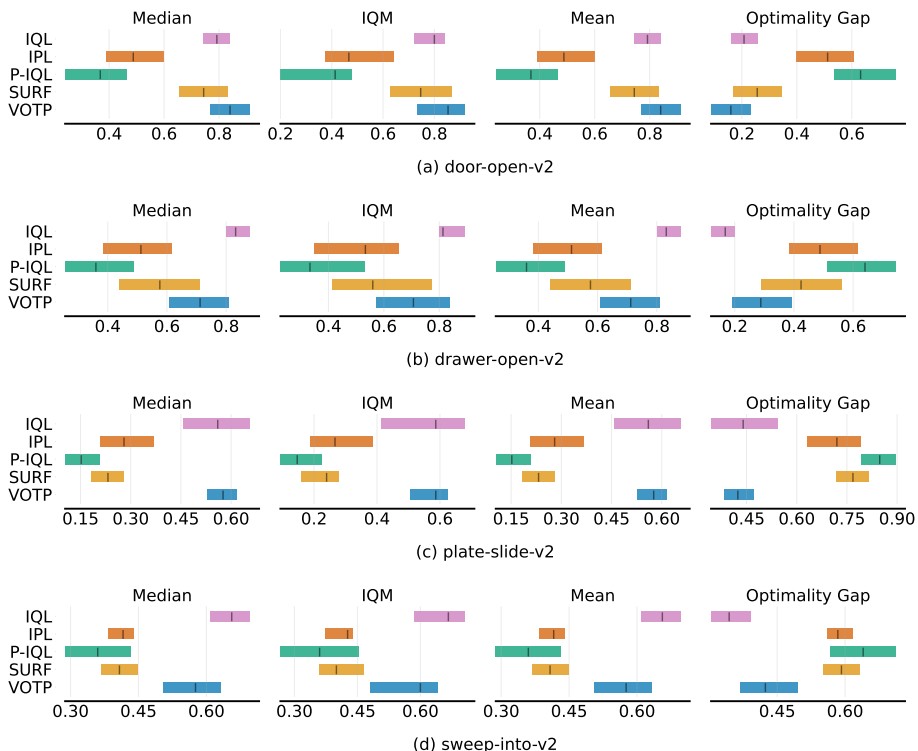

*Figure 13.* Aggregate metrics on MetaWorld manipulation tasks with 95% confidence intervals (CIs) across five runs. Higher mean, median and IQM scores and lower optimality gap are better. The CIs are estimated using the percentile bootstrap with stratified sampling.

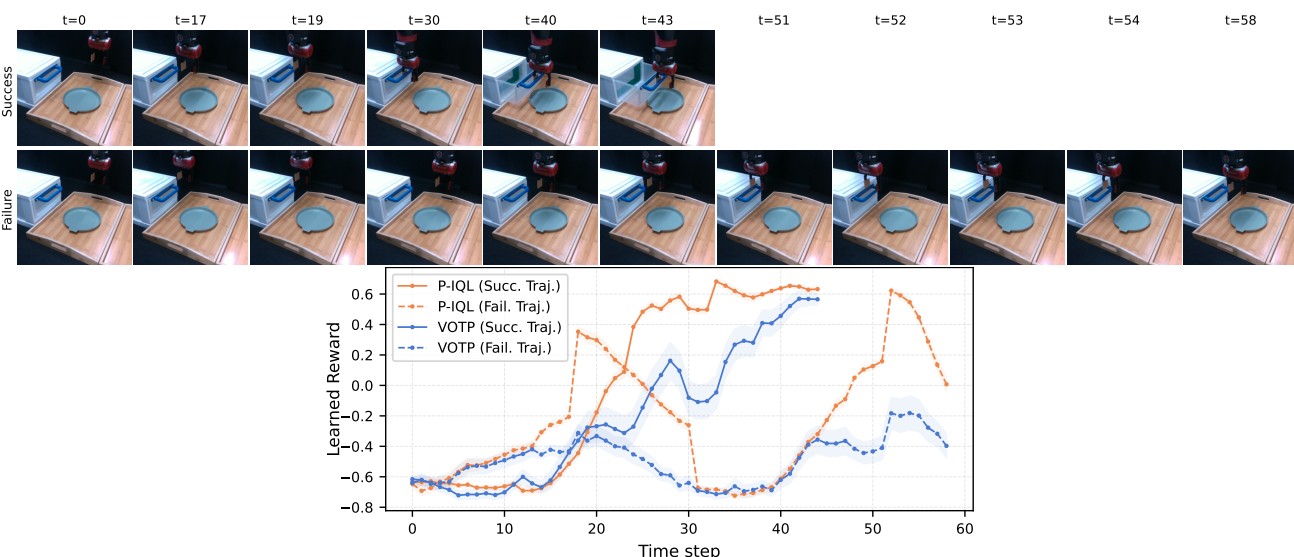

*Figure 14. Drawer Open*: Examples of successful and failed trajectories at each time step (top) with the corresponding reward outputs over timesteps from VOTP and P-IQL (bottom).

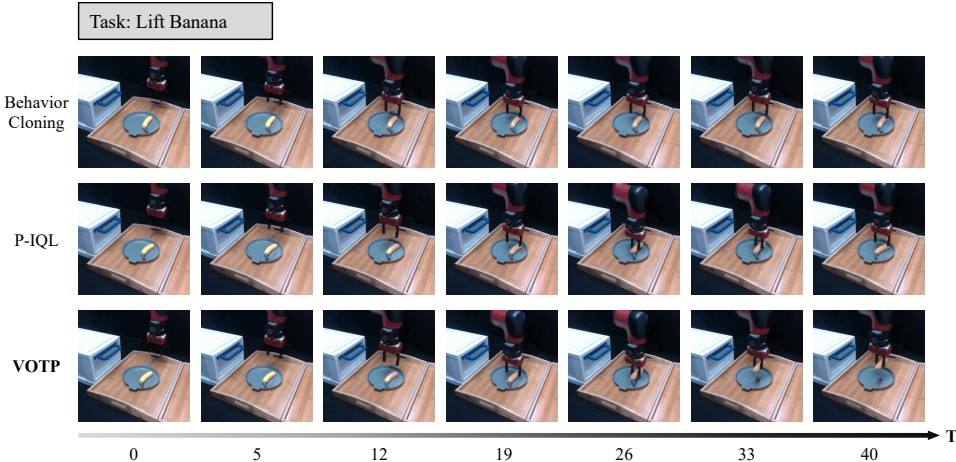

*Figure 15.* Snapshot of rollouts on *Lift Banana* task from BC, P-IQL and VOTP. The BC agent fails to descend to the banana and cannot grasp it. The P-IQL agent grasps the banana but does not lift and just release it. VOTP agent successfully reaches the banana, grasps it, and lifts it to a specified height.

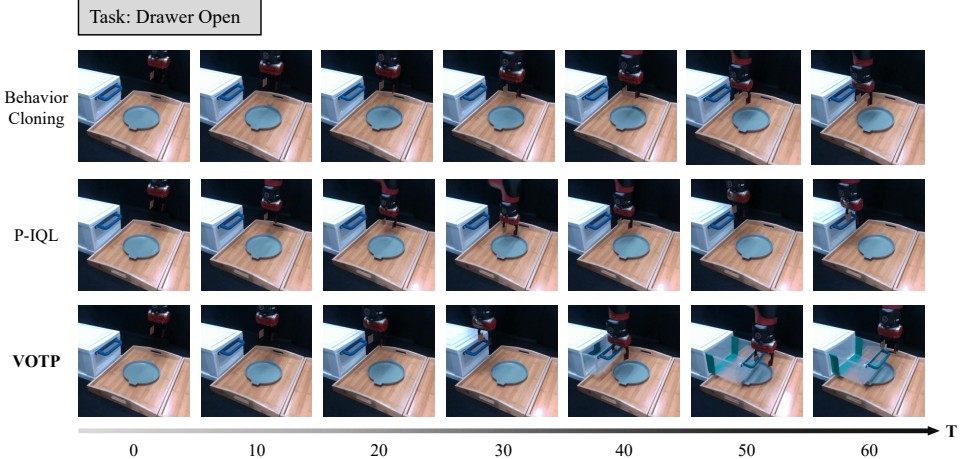

*Figure 16.* Snapshot of rollouts on *Drawer Open* task from BC, P-IQL and VOTP. In both BC and P-IQL, the agent barely reaches the handle after wandering and fails to pull the drawer open. VOTP agent, however, reaches the handle directly and pull it open successfully.

# C. Additional Results and Analysis

## C.1. Ablation Study on Cost Metric

To examine the sensitivity of VOTP to the choice of cost function, we evaluate it using Euclidean and cosine distances when computing the optimal transport plan. Table 9 shows that VOTP performs robustly under both choices.

*Table 9.* Performance of VOTP with different cost functions.

| Dataset | Euclidean | Cosine |
|---|---|---|
| hopper-medium-replay-v2 | 91.1 ±4.7 | 87.0 ±4.2 |
| walker2d-medium-replay-v2 | 66.3 ±5.6 | 64.1 ±14.3 |
| door-open | 84.0 ±8.4 | 82.0 ±8.7 |
| drawer-open | 71.2 ±11.7 | 75.8 ±4.2 |
| plate-slide | 57.6 ±5.4 | 57.0 ±12.4 |
| sweep-into | 57.6 ±7.4 | 58.0 ±11.8 |
| Average | 71.3 | 70.7 |

## C.2. RL Performance between GT Reward and Incorrect Rewards

For each dataset, we verify that RL performance differs when training with ground-truth (GT) rewards versus incorrect rewards (Choi et al., 2024). We use the three incorrect reward functions introduced in (Li et al., 2023): (1) Zero: all rewards are set to $r(s, a) = 0$; (2) Random: each transition takes a reward value sampled from a uniform distribution $U(0, 1)$; and (3) Negative: each transition's reward is replaced with the negation of the true reward, $-r(s, a)$. Following (Li et al., 2023), we report the performance of behavior cloning (BC), offline RL with GT rewards, and offline RL with incorrect rewards for each dataset in Table 10.

*Table 10.* Performance of IQL (Kostrikov et al., 2022) on each dataset under GT and incorrect rewards. Mean and standard deviation are computed over 5 random seeds.

| Dataset | BC | GT | Zero | Random | Negative |
|---|---|---|---|---|---|
| hopper-medium-replay-v2 | 33.7 ±8.5 | 87.5 ±7.4 | 28.0 ±8.3 | 48.2 ±6.7 | 0.6 ±0.0 |
| hopper-medium-expert-v2 | 55.1 ±2.8 | 104.5 ±4.5 | 53.6 ±1.4 | 67.8 ±9.8 | 16.8 ±5.5 |
| walker2d-medium-replay-v2 | 19.2 ±7.3 | 72.6 ±4.9 | 19.4 ±2.7 | 47.3 ±13.6 | 0.1 ±0.3 |
| walker2d-medium-expert-v2 | 100.4 ±13.4 | 109.9 ±0.5 | 100.7 ±7.7 | 69.7 ±7.1 | 20.9 ±0.7 |
| door-open | 57.5 ±3.0 | 79.2 ±5.9 | 59.6 ±2.1 | 52.0 ±2.8 | 39.0 ±7.7 |
| drawer-open | 61.5 ±3.7 | 83.2 ±4.7 | 61.0 ±1.7 | 59.8 ±3.3 | 46.8 ±5.5 |
| plate-slide | 39.1 ±2.5 | 56 ±11.9 | 38.4 ±2.0 | 34.0 ±10.8 | 24.0 ±5.7 |
| sweep-into | 49.3 ±2.1 | 65.6 ±5.4 | 46.6 ±2.6 | 48.8 ±4.7 | 27.0 ±9.5 |

## C.3. Computational Cost for Pseudo-Preference Genration

We train VOTP on an RTX 4090 GPU with 24 CPU cores (AMD Ryzen Threadripper 7960X). Feature extraction takes approximately 20 minutes for 50k segments. We compare sequential pseudo-labeling (labeling one pair at a time) with a parallel pseudo-labeling strategy (labeling 100 pairs concurrently). As shown in Table 11, the time required to generate pseudo-preference labels scales with the number of labels. We also find that the simple parallel technique effectively mitigates generation time, particularly when the number of labels is large. Policy training is implemented using IPL's source code [3] and takes roughly 1.5 hours for each run.

*Table 11.* Computational cost of VOTP for generating pseudo-preference labels for 10k unlabeled pairs under different dataset sizes.

| $N$ labels | 10 | 25 | 50 | 100 | 200 | 500 |
|---|---|---|---|---|---|---|
| Sequential time (minutes) | 0.15 | 0.2 | 0.75 | 3 | 12 | 60 |
| Parallel time (minutes) | - | - | - | 1 | 1.7 | 6.6 |

---

[3] https://github.com/jhejna/inverse-preference-learning

## C.4. Preference Learning from Human Teachers

We further evaluate VOTP using real human preference feedback across six datasets, with results summarized in Table 12. For D4RL locomotion, we use the human preference labels provided by Kim et al. (2023). For MetaWorld, we collect preferences from four non-robotic participants across four datasets. We observe a slight performance drop in *walker2d* but largely stable performance in the remaining environments.

*Table 12.* Performance of VOTP with preference feedbacks from human teachers. Mean and standard deviation are computed over 5 random seeds.

| Dataset | Scripted Teacher | Human Teacher |
|---|---|---|
| hopper-medium-expert-v2 | 105.7 $\pm$6.0 | 109.3 $\pm$1.7 |
| walker2d-medium-replay-v2 | 66.3 $\pm$5.6 | 59.4 $\pm$8.8 |
| walker2d-medium-expert-v2 | 108.1 $\pm$2.2 | 90.8 $\pm$7.0 |
| door-open | 84.0 $\pm$8.4 | 85.6 $\pm$8.6 |
| drawer-open | 71.2 $\pm$11.7 | 70.4 $\pm$6.5 |
| plate-slide | 57.6 $\pm$5.4 | 55.9 $\pm$11.3 |
| sweep-into | 57.6 $\pm$5.4 | 56.8 $\pm$11.3 |
| Average | 78.6 | 75.5 |

## C.5. Accuracy of Pseudo-preference Labels

*Table 13.* Accuracy of generated pseudo-labels: We calculate accuracy by comparing against ground-truth scripted preference labels (excluding equally preferred pairs). Overall, VOTP generates high-quality pseudo-labels with only a handful of labeled preference queries.

| Domain | Task | Accuracy (%) |
|---|---|---|
| D4RL Locomotion | hopper-medium-expert-v2 | 90.3 |
| | walker2d-medium-replay-v2 | 98.8 |
| | walker2d-medium-expert-v2 | 93.6 |
| MetaWorld Manipulation | door-open | 93.1 |
| | drawer-open | 97.4 |
| | plate-slide | 95.2 |
| | sweep-into | 67.0 |

## C.6. Additional Comparison of Learned Rewards with Ground-truth Rewards

We examine the reward values estimated by the learned reward models of P-IQL and VOTP. Figure 17 shows scatter plots of the estimated rewards of segments (*y*-axis) against the GT rewards (*x*-axis) for each dataset. As shown, the reward estimates produced by VOTP exhibit a significantly stronger correlation with GT rewards compared to those of P-IQL.

## C.7. Visualization for Pseudo-labeling Process

Qualitative results of the pseudo-labeling process on the *drawer-open* task in MetaWorld are shown in Figure 18. We present the labeled segments and their corresponding preference matrix, as well as the cost matrix, transport plans, and preference scores for four examples of unlabeled pairs. As shown, VOTP effectively aggregates preferences from labeled pairs to produce accurate pseudo-preferences for unlabeled pairs.

## C.8. Discussion of Uniformity Assumption

In our framework, the assumption of uniform marginals (Eq. 4) is naturally motivated by the data-loading process. Since trajectory segments are sampled uniformly from the offline dataset without replacement, each segment appears with equal frequency during the pseudo-labeling phase. Consequently, the empirical distribution over the sampled segments is effectively uniform, making Optimal Transport with uniform marginals a principled and appropriate relaxation of the matching problem.

## D. Extended Discussion on Broader Context and Future Directions

While our framework establishes a scalable foundation for preference alignment, our current evaluations focus on standard Gaussian policies. However, recent advances in generative modeling (Ho et al., 2020; Lipman et al., 2022; Pham et al., 2025; Ton et al., 2025) have significantly expanded the capacity to model complex and multimodal action distributions in both reinforcement learning (Lillicrap et al., 2016; Haarnoja et al., 2018; Luu & Yoo, 2021; Nguyen et al., 2021) and imitation learning (Chen et al., 2021; Luu et al., 2024). In particular, continuous-time generative frameworks such as flow matching have emerged as highly expressive policy representations (Park et al., 2025; Nguyen & Yoo, 2026; Nguyen et al., 2026) and are increasingly being integrated into large-scale vision-language-action (VLA) architectures (Chi et al., 2023; Black et al., 2024; Bjorck et al., 2025). Extending VOTP to align these powerful generative policies therefore represents a promising direction for future research (Xia et al., 2025; Miao et al., 2025). Specifically, our framework could leverage optimal transport to efficiently fine-tune foundation-scale VLAs without requiring expensive dense feedback annotations. Beyond policy parameterization, the effectiveness of preference propagation also depends on representation quality for capturing nuanced behaviors in precision manipulation. Incorporating recent advances in self-supervised pretraining (Oquab et al., 2023; Nguyen et al., 2023; Assran et al., 2023) and 3D scene understanding (Vu et al., 2022; 2023; Park et al., 2026) into the VOTP latent space could further improve semantic structure and generalization. Finally, although our experiments focus on robotics and continuous control, the underlying principle of optimal-transport-based preference propagation may generalize to other domains characterized by scarce annotations and structured sequential data, including healthcare diagnostics, assistive systems, and decision-making pipelines (Kim et al., 2021; Yoon et al., 2022; 2023; Koo et al., 2024; 2025).

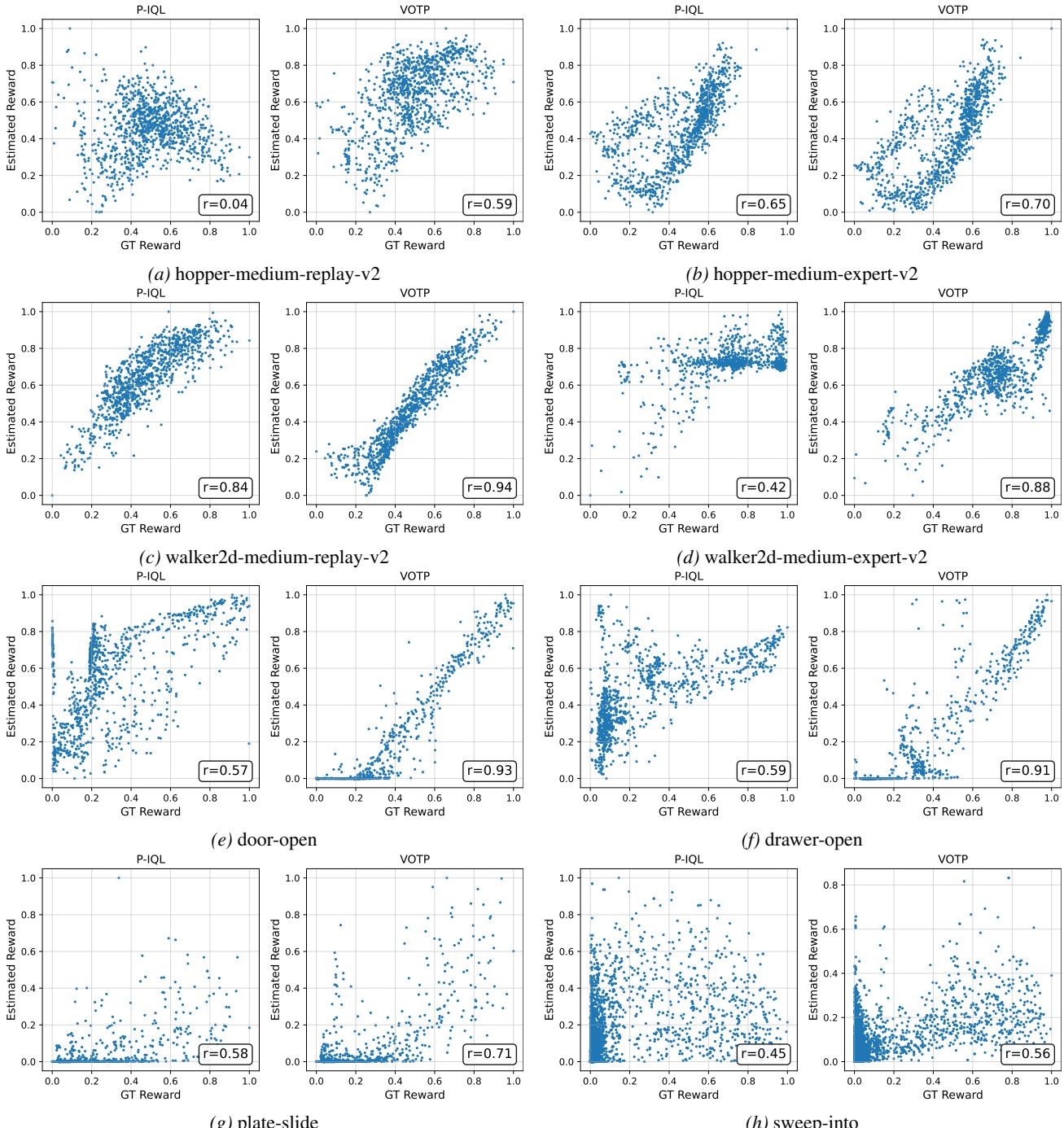

*Figure 17.* Correlation between learned rewards and ground-truth rewards for P-IQL and VOTP. Pearson correlation coefficients ($r$) are shown for each dataset.

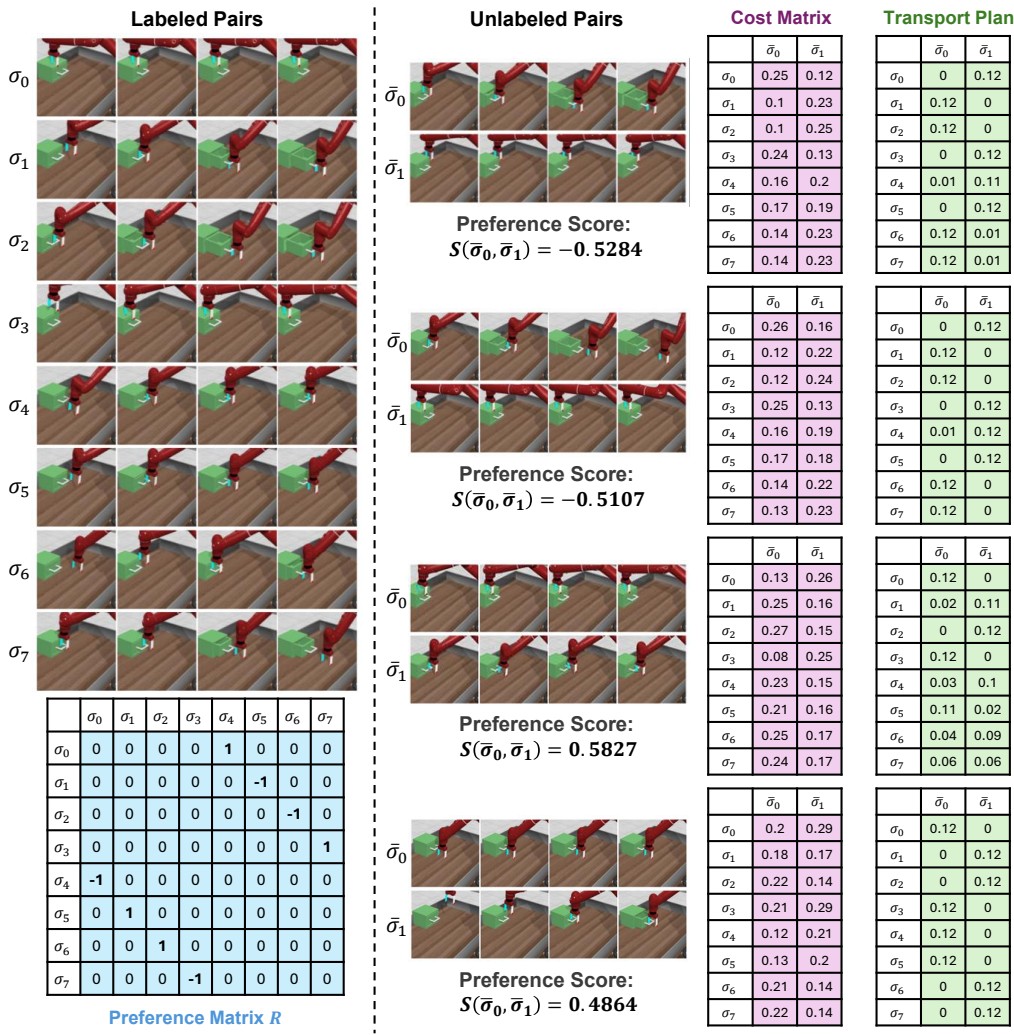

*Figure 18.* Additional visualizations of the pseudo-labeling process from VOTP on the drawer-open task. We use 4 labeled pairs. All cost matrix and transport plan entries are rounded to two decimals. For clarity, each segment (originally 64 frames) is uniformly downsampled to 4 frames for visualization.

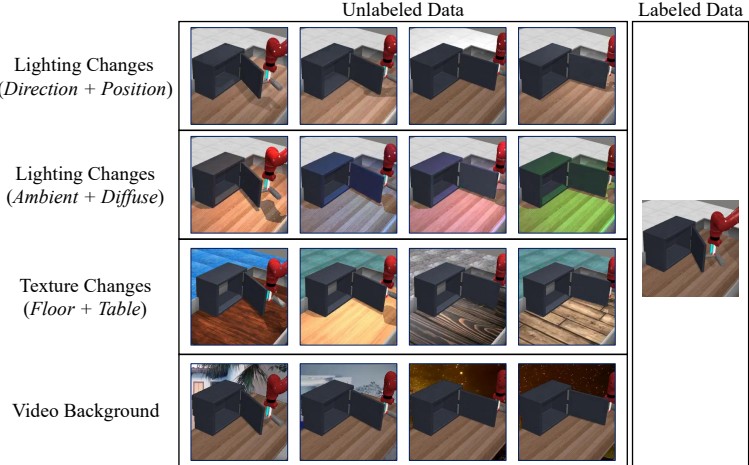

*Figure 19.* Examples of scenarios with different types of visual distractions. For lighting changes, we consider two distractors: (1) variations in the direction and position of the light source, and (2) variations in ambient and diffuse lighting. For texture changes, we randomize the textures of the table and floor (20 possible combinations). For video background distractions, we use both the easy and hard videos from Yuan et al. (2023). Example segment clips are available at this link. Note that the distractor variation within each segment is fixed, but may vary across segments.

