# OpenReview forum: "Video-Based Optimal Transport for Feedback-Efficient Offline Preference-Based Reinforcement Learning"
_ICML.cc/2026/Conference — ICML 2026 spotlight_

### Official Review · Reviewer_mGZK · 2026-02-14

**Soundness:** 3
**Presentation:** 3
**Significance:** 3
**Originality:** 3
**Overall Recommendation:** 4
**Confidence:** 3

**Summary:**

This paper studies feedback-efficient offline preference-based reinforcement learning (PbRL) under limited human labeling budgets. The proposed method, Video-based Optimal Transport Preference (VOTP), leverages pretrained video foundation model (ViFM) representations and optimal transport to align labeled and unlabeled visual trajectory segments, producing pseudo-preference labels at scale. These pseudo-labels augment a small set of labeled comparisons to train a reward model, enabling improved downstream offline RL across locomotion and manipulation benchmarks, with additional robustness tests under visual distractors and a small real-robot evaluation.

**Compliance With Llm Reviewing Policy:**

Affirmed.

**Final Justification:**

The paper addresses an important problem in feedback-efficient offline preference-based RL, and I found the proposed use of video foundation model representations and optimal transport for pseudo-labeling both clear and well motivated. The empirical results, including the real-robot experiments, support the practical value of the approach.

My main concerns were about the robustness of pseudo-labeling under sparse, noisy, or biased preferences. The authors’ rebuttal addressed these concerns satisfactorily and clarified the relevant limitations. Overall, the rebuttal strengthened my confidence in the paper and reinforced my prior positive assessment, so I maintain my current recommendation.

**Key Questions For Authors:**

1. How sensitive is VOTP to the choice of ViFM (e.g., S3D vs other video encoders) and to the OT cost function (Euclidean vs alternatives)? Can you provide guidance for practitioners?
2. How does VOTP behave when the labeled preference set contains inconsistent or noisy human labels (e.g., disagreement across annotators)?
3. Do you observe cases where OT alignment produces confident but wrong pseudo-labels (e.g., visually similar but semantically different behaviors), and can uncertainty be estimated to filter them?

**Limitations:**

yes

**Strengths And Weaknesses:**

Strengths:
- Clear and well-motivated approach to semi-supervised PbRL: unlabeled segment pairs are free in offline datasets, and exploiting them directly addresses labeling cost.
- The OT-based pseudo-labeling is conceptually clean and appears broadly applicable to vision-based PbRL without requiring task-specific heuristics.
- Real-robot results (few labels, clear success-rate gains over BC and P-IQL) help validate practical utility.

Weaknesses:
- Most large-scale benchmark results use scripted teachers / ground-truth rewards to generate preferences, which may not reflect real human preference noise, inconsistency, or shifting criteria.
- It is unclear how robust pseudo-labeling is when labeled preferences are sparse but also biased (e.g., concentrated on a narrow behavior region), since OT alignment could propagate that bias.

---

> ### Author Rebuttal · Authors · 2026-03-30
>
> We thank the reviewer for their detailed and thoughtful review, and for recognizing the clear motivation and the practical utility of our approach. We address the questions in detail below.
>
> - **Comment 1**. It is unclear how robust pseudo-labeling is when labeled preferences are sparse but also biased, since OT alignment could propagate that bias.
>
>   **Response 1**. We agree with the reviewer's intuition. Our OT-based pseudo-labeling essentially anchors unlabeled segments to known preferred behaviors. Therefore, a diverse set of preferred behaviors enables the extraction of diverse pseudo-labels, facilitating reward model generalization. Conversely, if the initial preference support is highly biased toward a narrow behavioral region, VOTP may propagate this bias, leading to a less generalized reward model. While this challenge is common across most reward learning methods in offline PbRL, it can be effectively addressed in practice. Since offline datasets often contain a wide variety of behaviors, slightly expanding the labeled set naturally overcomes this bias, as shown in Figure 4. Also, active sampling techniques [2, 3] can be used to further encourage diversity.
>
> - **Q1**. How sensitive is VOTP to the choice of ViFM and to the OT cost function? Can you provide guidance for practitioners?
>
>   **A1**. As shown in Figure 3, we observe that the variation in performance across ViFMs is relatively smaller in manipulation tasks than in locomotion tasks, suggesting that sensitivity to ViFMs depends on the task domain. We also find that VOTP is robust to the choice of OT cost function, as detailed in Table 9. In practice, we recommend performing a simple check by training the reward model with different design choices and then qualitatively comparing its outputs on known successful and failed trajectories (as in our real-world experiments). This provides a simple and efficient way to inspect the behavior of the reward model before committing to full policy training or environment rollouts.
>
> - **Q2**. How does VOTP behave when the labeled preference set contains inconsistent or noisy human labels?
>
>   **A2**. Our experiments using human preferences are detailed in Table 12. We observe a slight performance drop on the walker2d dataset, while performance remains largely stable across the others. To investigate this, we conducted an additional evaluation measuring classification metrics (e.g., F1 score, accuracy, etc.) between pseudo-labels and human preferences on datasets with a large number of labels available in [1]. We find that pseudo-labeling for walker2d is more challenging, with metrics lower than those for hopper. Also, in walker2d-medium-replay, the accuracy of pseudo-labels generated from human preferences is lower than that obtained using a scripted teacher (in Table 13),  which we believe accounts for the observed performance drop. We kindly refer the reviewer to **A1** in our response to Reviewer XRLr for the table results.
>
> - **Q3**. Do you observe cases where OT alignment produces confident but wrong pseudo-labels and can uncertainty be estimated to filter them?
>
>   **A3**. Yes, this can occasionally occur, particularly when two segments exhibit visually similar but slightly different behaviors. For instance, we manually examined incorrectly pseudo-labeled pairs with high preference scores in the door-open task and observed that in most cases, the door remained closed in both segments, while the robot arm made slight, varied movements around the handle or on top of the door. While these cases should ideally be labeled as equally preferred, VOTP assigned a highly confident preference to one of the segments. We conjecture that this occurs when some of these movements align more closely with a preferred behavior in the ViFM latent space. We observed a similar phenomenon in the drawer-open task.
>
>   Importantly, we observed no cases of "catastrophic" mislabeling; that is, we found no instances where a completely failed segment is confidently preferred over a successful one. Instead, the errors primarily arise from misclassifying "equal" pairs as having a preference (i.e., label 0.5 $\rightarrow$ {0, 1}), rather than flipping the direction of a true preference (i.e., 0 $\rightarrow$ 1 or 1 $\rightarrow$ 0). As a result, the induced noise is substantially less harmful for downstream reward learning. To mitigate these ambiguous pairs, one could leverage a reward model's uncertainty [2] (e.g., calculating the variance across an ensemble's preference predictions) or apply active sampling techniques [3] to filter them out. We leave this exploration for future work.
>
> [1] Kim, C., et al. "Preference transformer: Modeling human preferences using transformers for rl." ICLR (2023).\
> [2] Shin, D, et al. "Benchmarks and algorithms for offline preference-based reward learning." TMLR (2023).\
> [3] Mu, Ni, et al. "Clarify: Contrastive preference reinforcement learning for untangling ambiguous queries." ICML (2025).

---

> > ### Author Rebuttal · Reviewer_mGZK · 2026-04-01
> >
> > The rebuttal satisfactorily addresses my main concerns and strengthens my confidence in the paper, so I will maintain my current score.

---

### Official Review · Reviewer_uxwd · 2026-02-27

**Soundness:** 4
**Presentation:** 4
**Significance:** 4
**Originality:** 3
**Overall Recommendation:** 6
**Confidence:** 4

**Summary:**

This paper proposes VOTP, a semi-supervised reward-modeling framework that reduces labeling cost by generating high-quality pseudo-labels for unlabeled video segments. VOTP uses optimal transport and features from video foundation models to compute preference scores for video pairs, leveraging a small labeled set to supervise the unlabeled data. Experiments on two standard benchmarks and a real-world manipulation task demonstrate the effectiveness of the proposed approach.

**Compliance With Llm Reviewing Policy:**

Affirmed.

**Key Questions For Authors:**

1. How does the trajectory length $H$ affect VOTP’s performance? How should a proper $H$ be chosen for real-world tasks?
2. How are features for a video segment computed using image foundation models (right column, Line 297)? Are features extracted for all frames in the segment and then summarized?
3. In Fig. 5, the authors report the score and success rate obtained by VOTP. However, for some tasks (e.g., walker2d and drawer-open), there is no clear performance drop at the maximum $\tau$ values shown. Would the score decrease if $\tau$ were increased beyond the plotted maxima (i.e., beyond 0.2 and 0.5 for these two tasks)?
4. In the robustness experiments (Sec. 5.4), are the training settings the same as those used in Table 1 (e.g., number of training rounds, $\tau$ selection, and $N$ selection)?
5. Why retaining only those with high scores could mitigate the impact of inaccurate pseudo-labels (Line 212, right column)?

**Limitations:**

yes

**Strengths And Weaknesses:**

**Strengths**
1.  This paper is well organized and easy to follow. The overview in Fig. 1, together with an example, helps readers understand the workflow of VOTP.
2. The idea of inferring preferences from labeled video segments using optimal transport is clearly presented and intuitive.
3. The experimental results demonstrate the core contribution of VOTP, its effectiveness compared to other methods, and the impact of key hyperparameters. Reporting standard deviation makes the results more convincing.
4. The real-world application makes the method practical.

**Weaknesses**

No major weaknesses. As a small suggestion, this work would be more impactful and practical if the authors could demonstrate VOTP on more challenging real-world tasks (e.g., cloth folding), where rewards are difficult to obtain through reward engineering, in future work.

I would like to increase my score if the authors could open-source the VOTP implementation.

---

> ### Author Rebuttal · Authors · 2026-03-30
>
> We thank the reviewer for their in-depth review and for recognizing the clear motivation, clear presentation, and strong experimental results of VOTP. Encouraged by your suggestion, we have open-sourced a standalone, anonymous implementation of our core labeling mechanism to facilitate future research: https://anonymous.4open.science/r/votp-3CE8. We address your remaining questions in detail below.
>
> - **Q1**. How does the trajectory length $H$ affect VOTP’s performance? How should a proper $H$ be chosen for real-world tasks?
>
>   **A1**. In our experiments, we find that VOTP is robust to the choice of $H$, with values of 100 for D4RL, 64 for MetaWorld, and 16 for real-world tasks. In practice, depending on the robot’s control frequency and the specific video encoder used, we recommend starting with the smallest frame window supported by the encoder (e.g., a minimum of 16 frames for S3D). For setups with high control frequencies, segments can be downsampled by skipping frames to reduce redundancy.
>
> - **Q2**. How are features for a video segment computed using image foundation models (right column, Line 297)? Are features extracted for all frames in the segment and then summarized?
>
>   **A2**. When using image foundation models, features are extracted for each frame in the segment. The distance between two segments is then computed as the average pairwise distance between these frame-level features.
>
> - **Q3**. In Fig. 5, the authors report the score and success rate obtained by VOTP. However, for some tasks (e.g., walker2d and drawer-open), there is no clear performance drop at the maximum $\tau$ values shown. Would the score decrease if $\tau$ were increased beyond the plotted maxima (i.e., beyond 0.2 and 0.5 for these two tasks)?
>
>   **A3**. Yes, the performance does indeed decrease as $\tau$ increases further. We provide extended results showing this performance drop when additionally increasing the value of $\tau$ at this [Link](https://anonymous.4open.science/r/votp-3CE8/figures/ID-25218-rebuttal.pdf).
>
> - **Q4**. In the robustness experiments (Sec. 5.4), are the training settings the same as those used in Table 1 (e.g., number of training rounds, $\tau$ selection, and $N$ selection)?
>
>   **A4**. Yes, the training settings in the robustness experiments are identical to those used in Table 1, with the sole exception of the preference threshold $\tau$. The best-performing value for $\tau$ varies depending on the type of visual distraction.
>
> - **Q5**. Why retaining only those with high scores could mitigate the impact of inaccurate pseudo-labels (Line 212, right column)?
>
>   **A5**. After thresholding the preference scores via Equation (7), less confident pairs are assigned an "equally preferable" label (i.e., $\tilde{y} = 0$). While these pairs could, in principle, be used for reward learning, they consist of a mix of (1) genuinely equally preferred pairs and (2) pairs that have a true preference but are assigned a tie due to low confidence. Because type (2) pairs introduce harmful noise into the reward learning process, we exclude these pairs entirely, thereby mitigating the impact of inaccurate pseudo-labels.

---

> > ### Author Rebuttal · Reviewer_uxwd · 2026-04-01
> >
> > Thanks for the authors’ response. I am increasing my score to 6 in recognition of the paper’s potential contribution to researchers in RL and embodied AI.

---

### Official Review · Reviewer_XRLr · 2026-03-11

**Soundness:** 4
**Presentation:** 4
**Significance:** 4
**Originality:** 4
**Overall Recommendation:** 5
**Confidence:** 4

**Summary:**

This paper addresses the challenge of Preference-based Reinforcement Learning by leveraging advanced video foundation models. The authors propose Video-based Optimal Transport Preference (VOTP), a semi-supervised framework designed to generate high-fidelity pseudo-preference labels for large volumes of unlabeled data. This is achieved by aligning visual trajectories within the feature space of a pre-trained video foundation model using Optimal Transport. The authors demonstrate the effectiveness of their approach across two simulation benchmarks and one real-world robotic experiment.

**Compliance With Llm Reviewing Policy:**

Affirmed.

**Final Justification:**

I appreciate the authors for providing additional experimental results during the rebuttal phase. Overall, I recommend acceptance.

**Key Questions For Authors:**

1. The entire framework relies heavily on the assumption that the generated pseudo-preference labels are of "high fidelity." However, this claim lacks direct quantitative validation. Could the authors explicitly report the Precision, Recall, and F1 score of the predicted preferences? I strongly recommend evaluating and presenting these classification metrics on a held-out dataset where ground-truth human preference labels are available. This analysis is crucial to understanding the noise tolerance and reliability of the VOTP labeling mechanism.

2. Given that the authors collected 50 expert demonstrations for the real-world experiments, the necessity of mapping these to a preference-based framework becomes questionable. Why not directly employ Optimal Transport (OT) reward learning methods, such as ROT[1], which directly learn reward functions by comparing the distribution of expert demonstrations with agent rollout trajectories? I request that the authors discuss the theoretical or empirical advantages of their semi-supervised preference approach over direct OT reward learning, and ideally, include ROT (or a similar method) as a baseline comparison.

[1] Watch and match: Supercharging imitation with regularized optimal transport.

**Limitations:**

yes

**Strengths And Weaknesses:**

# Strengths:
The idea of utilizing the rich, pre-trained representations of video foundation models for aligning trajectories and generating pseudo-preferences in a semi-supervised manner is highly motivated and intuitively sound. The empirical results across multiple simulation benchmarks are solid. Furthermore, the inclusion of real-world robotic experiments significantly strengthens the paper's practical relevance.

# Weaknesses:
Despite the promising results, my primary concerns lie in the empirical validation of the core mechanism, specifically the reliability of the generated pseudo-labels and the absence of critical baselines that directly utilize demonstration data for reward learning. Please refer to the detailed questions below.

---

> ### Author Rebuttal · Authors · 2026-03-30
>
> We thank the reviewer for their insightful and constructive feedback, and for recognizing the motivation, empirical strengths, and practical relevance of our approach. We address the comments below.
>
> - **Q1**. The entire framework relies heavily on the assumption that the generated pseudo-preference labels are of "high fidelity." However, this claim lacks direct quantitative validation. Could the authors explicitly report the Precision, Recall, and F1 score of the predicted preferences? I strongly recommend evaluating and presenting these classification metrics on a held-out dataset where ground-truth human preference labels are available. This analysis is crucial to understanding the noise tolerance and reliability of the VOTP labeling mechanism.
>
>   **A1**. We evaluated classification metrics for our generated pseudo-preference labels against ground-truth human preferences using the datasets provided in [1]. The results are shown in the table below. We randomly sampled labeled sets, generated pseudo-labels, and computed the metrics over 20 trials for the *hopper-medium-replay* and *walker2d-medium-replay* datasets (each containing 500 ground-truth labels).
>
>   |                           | Precision  | Recall         | F1 score    | Accuracy   |
>   | ---                       | ---        | ---            | ---         |---         |
>   | hopper-medium-replay | 90.3 ± 5.9 | 81.9 ± 7.0     | 85.9 ± 6.2  | 83.7 ± 7.3 |
>   | walker2d-medium-replay | 81.1 ± 9.2 | 75.2 ± 10.5    | 78.0 ± 9.8  | 76.2 ± 9.9 |
>
>   These results demonstrate that our labeling mechanism remains relatively reliable even in the presence of potentially noisy and inconsistent human preference labels. Furthermore, the lower metrics on the *walker2d* dataset align with and help explain the performance drop observed on *walker2d* in Table 12 (Appendix) when using human labels.
>
>
> - **Q2**. Given that the authors collected 50 expert demonstrations for the real-world experiments, the necessity of mapping these to a preference-based framework becomes questionable. Why not directly employ Optimal Transport (OT) reward learning methods, such as ROT [2], which directly learn reward functions by comparing the distribution of expert demonstrations with agent rollout trajectories? I request that the authors discuss the theoretical or empirical advantages of their semi-supervised preference approach over direct OT reward learning, and ideally, include ROT (or a similar method) as a baseline comparison.
>
>   **A2**. In this work, we focus on the offline RL setting, which assumes access to a pre-collected, suboptimal dataset. To simulate this in our real-world experiments, we intentionally collected a suboptimal dataset (averaging a 50% success rate), rather than full expert demonstrations. While an OT-based reward approach (essentially inverse RL) could be applied in this setting, it would require collecting additional expert demonstrations. In contrast, our preference learning method only requires human feedback on segment pairs drawn directly from the existing offline dataset, thereby avoiding the need to collect extra expert demonstrations.
>
>   Ultimately, both approaches aim to estimate a reward function that is then used by an offline RL algorithm to learn a policy. To empirically investigate the effectiveness of these reward functions, we conducted experiments using ROT-based rewards on four MetaWorld tasks. We obtained the ROT rewards by following the procedure in [2] (i.e., using demonstrations generated by a scripted expert policy provided by the MetaWorld benchmark), while keeping all other training settings identical (e.g., IQL hyperparameters). The results in the table below indicate that preference learning yields better policy performance in this offline setting. We believe that OT-based reward learning is more beneficial when the agent is allowed to interact more with the environment, as observed in [2, 3, 4, 5].
>
>   |         | door-open   | drawer-open | plate-slide | sweep-into |
>   | ---     | ---         | ---         | ---         | ---        |
>   | VOTP    | 84.0 ± 8.4  | 71.2 ± 11.7 | 57.6 ± 5.4  | 57.6 ± 7.4 |
>   | ROT     | 63.2 ± 5.9  | 60.0 ± 16.8 | 34.4 ± 6.5  | 51.2 ± 5.9 |
>
>
> [1] Kim, C., et al. "Preference transformer: Modeling human preferences using transformers for rl." ICLR (2023).\
> [2] Haldar, Siddhant, et al. "Watch and match: Supercharging imitation with regularized optimal transport." CoRL (2023).\
> [3] Fu, Yuwei, et al. "Robot policy learning with temporal optimal transport reward." NeurIPS (2024).\
> [4] Haldar, Siddhant, et al. "Teach a robot to fish: Versatile imitation from one minute of demonstrations." RSS (2023).\
> [5] Huey, William, et al. "Imitation learning from a single temporally misaligned video." ICML (2025).

---

> > ### Author Rebuttal · Reviewer_XRLr · 2026-04-02
> >
> > Thanks for your response. My concerns have been resolved. I will keep my score and recommend accepting this paper.

---

### Official Review · Reviewer_r8yx · 2026-03-13

**Soundness:** 2
**Presentation:** 3
**Significance:** 3
**Originality:** 3
**Overall Recommendation:** 5
**Confidence:** 3

**Summary:**

This paper proposes Video-based Optimal Transport Preference labeling (VOTP), a semi-supervised approach for offline preference-based RL that aims to reduce the number of human preference labels needed for reward learning. The core idea is to leverage Video Foundation Models (ViFMs) to embed video trajectory segments, then use optimal transport (OT) to align labeled and unlabeled segment pairs in the ViFM latent space, enabling pseudo-preference label inference for unlabeled pairs from only a small labeled set. The inferred pseudo-labeled pairs are combined with labeled preferences to train a Bradley-Terry style reward model, and the offline dataset is then re-labeled with the learned reward for downstream offline RL (IQL). The paper evaluates on D4RL locomotion and MetaWorld manipulation, as well as a real-world robotic arm task.

**Compliance With Llm Reviewing Policy:**

Affirmed.

**Final Justification:**

The authors have adequately addressed my concerns in rebuttal.

**Key Questions For Authors:**

1. Could you provide a clearer compute efficiency comparison against the baselines, and show the impact of scaling with the number of labeled pairs with empirical results?
2. What do you believe explains the gap between scripted-teacher and human-teacher results on some tasks, particularly the walker2d?

**Limitations:**

yes

**Strengths And Weaknesses:**

Strengths:
1. In this paper's experiments, VOTP is shown to achieve competitive performance with as few as 10 labeled preference pairs in several tasks, a large improvement over standard Preference-based RL methods.
2. By leveraging video foundation models, VOTP captures temporal dynamics and subtle motion cues, and the paper’s ablation shows these video encoders generally outperform image foundation models within the VOTP framework. The paper also presents experiments showing that VOTP achieves robustness to a variety of nuisance variations.
3. The real robotics arm experiment further confirms the the method's feasibility in real-world scenarios.

Weaknesses:
1. A limitation is VOTP's additional computational overhead from OT-based pseudo-label generation, whose cost grows with the number of labeled preference pairs. The paper reports this overhead is manageable in its experiments, but does not directly compare runtime or memory against baselines, so scalability to larger labeled datasets remains as a concern.
2. The real robotics arm experiment results are promising but still limited in scope. The paper evaluates only two tasks (Lift Banana and Drawer Open), using just 5 and 10 preference labels respectively.
3. The main experiment results rely on scripted preference labels rather than real human feedback. Although the appendix (table 12) includes a human-teacher evaluation, it shows noticeable drops on some tasks, but the paper does not discuss the source of this discrepancy or whether it stems from a mismatch between human preferences and the scripted label-generation process.

---

> ### Author Rebuttal · Authors · 2026-03-30
>
> We thank the reviewer for the detailed and thoughtful review, and for recognizing the effectiveness of our method, our strong empirical results, and our real-world robotic experiments. We address the questions in detail below.
>
> - **Q1**. Could you provide a clearer compute efficiency comparison against the baselines, and show the impact of scaling with the number of labeled pairs with empirical results?
>
>   **A1**. The table below reports training time and memory consumption for a single run on D4RL and MetaWorld. For fairness, all methods were evaluated on a single RTX 4090 GPU. While VOTP (using 10 labels, as in the main paper) incurs a slightly higher training cost than other reward learning methods due to ViFM encoding, it remains competitive and is significantly faster than FTB, which similarly leverages unlabeled data to enhance reward learning.
>
>   |       | CPL          | DPPO     | IPL/PIQL/SURF/LiRE   | FTB        | VOTP |
>   | ---       | ---          | ---      | ---                     | ---        | --- |
>   | D4RL      | 0.5h / 0.8GB | 1h / 1GB | 1.7h / 1GB              | 39h / 23GB  | 1.9h / 2GB |
>   | MetaWorld | 0.5h / 0.8GB | 1h / 1GB | 1h   / 1GB              | 16h / 8GB | 1.1h / 2GB |
>
>   Regarding scalability, Table 11 in the Appendix details the time required for pseudo-label generation (for 10k unlabeled pairs) as the number of labeled pairs ($N$) increases. In our framework, the primary computational bottleneck is the generation time associated with a large number of labeled preference pairs. To directly address the concern about scaling to larger labeled datasets, we implemented a parallel pseudo-labeling strategy that processes 100 pairs concurrently (originally, pseudo-labeling is performed sequentially, i.e., one sample at a time). The resulting generation times are shown in the table below, where the first row matches Table 11. We find that this simple technique effectively mitigates the generation time, particularly when the number of labels is large.
>
>     | $N$ labels  | 100     | 200     | 500     |
>     | ---         |  ---    | ---     | ---     |
>     | Sequential  | 3 min   | 12 min  | 60 min  |
>     | Parallel    | 1 min   | 1.7 min | 6.6 min |
>
> - **Q2**. What do you believe explains the gap between scripted-teacher and human-teacher results on some tasks, particularly the walker2d?
>
>   **A2**. We conjecture that the performance drop observed when learning from human-teacher, particularly in the *walker2d* datasets, stems from inherent noise and inconsistencies in the human labels, which in turn affect the quality of the generated pseudo-labels. To examine this, we conducted an additional evaluation measuring the precision, recall, F1 score, and accuracy of our pseudo-labels against ground-truth human preferences on the *hopper-medium-replay* and *walker2d-medium-replay* datasets. We find that pseudo-labeling for *walker2d-medium-replay* is indeed more challenging and and yields lower classification metrics than *hopper-medium-replay*. Additionally, in *walker2d-medium-replay*, the accuracy of pseudo-labels generated from human teacher preferences is lower than that obtained using a scripted teacher (shown in Table 13). This helps explain the observed performance drop. We kindly refer the reviewer to **A1** in our response to Reviewer XRLr for the quantitative results.

---

> > ### Author Rebuttal · Reviewer_r8yx · 2026-04-01
> >
> > Thank you for your reply! I will increase my score to 5.

---

### Decision · Program_Chairs · 2026-04-30

**Decision:**

Accept (spotlight)

**Comment:**

This paper studies feedback-efficient offline preference-based reinforcement learning (PbRL) under limited human labeling budgets. The proposed method, Video-based Optimal Transport Preference (VOTP), leverages pretrained video foundation model (ViFM) representations and optimal transport to align labeled and unlabeled visual trajectory segments, producing pseudo-preference labels at scale. These pseudo-labels augment a small set of labeled comparisons to train a reward model, enabling improved downstream offline RL across locomotion and manipulation benchmarks in simulation, with additional robustness tests under visual distractors and a small real-robot evaluation.

Based on the reviewer evaluations, the paper addresses an important research question which is well-motivated (XRLr, mGZK), well-presented (uxwd, mGZK). The experiments are technically sound (XRLr, uxwd) and demonstrate a large improvement over standard Preference-based RL methods (r8yx) for a broadly applicable approach (mGZK). VOTP also demonstrates robustness to a variety of nuisance variations (r8yx). All reviewers appreciate the real-world robotic experiments which significantly strengthens the paper's practical relevance (r8yx, XRLr, uxwd, mGZK)

The authors present new evaluations during the rebuttal to highlight the primary computational bottleneck of their approach and a parallel pseudo labeling mitigation strategy (r8yx) . They also show empirically better policy performance over a new baseline ROT (XRLr). They clarify several hyperparameters, experimental design choices, and results (uxwd).

They further explain their use of scripted preference labels for the main results and discuss the inherent noise and inconsistencies in human labels for certain datasets (r8yx). They also show that their labeling mechanism remains relatively reliable even in the presence of potentially sparse, noisy, inconsistent, or biased human preferences (XRLr, mGZK).

A recommendation from reviewer uxwd to further improve the paper for future work is to demonstrate VOTP on more challenging real-world tasks (e.g., cloth folding), where rewards are difficult to obtain through reward engineering. There is also a request to the authors to consider open-sourcing the VOTP implementation for the benefit of the research community.

Overall, all reviewers expressed that their concerns were addressed appropriately by the authors during the discussion period. Given all the reviews and discussions, I recommend acceptance.